# MINI-OMNI-REASONER: TOKEN-LEVEL THINKING-IN-SPEAKING IN LARGE SPEECH MODELS

## ABSTRACT

Reasoning is essential for effective communication and decision-making. While recent advances have equipped large language models (LLMs) and multimodal models (MLLMs) with strong reasoning abilities in text and vision-language tasks, integrating reasoning into large speech models (LSMs) remains at a preliminary stage. Existing methods typically adopt the thinking-before-speaking paradigm from textual models, where responses are delayed until reasoning is complete. This sequential formulation introduces significant latency, hindering real-time interaction. We propose Mini-Omni-Reasoner, a framework that enables reasoning in speech through a novel Thinking-in-Speaking formulation. By interleaving silent reasoning tokens with spoken response tokens at the token level, the model achieves continuous speech generation while maintaining logical consistency. To support this, we construct Spoken-Math-Problems-3M, a large-scale dataset designed for interleaved reasoning and response, ensuring that each spoken token is grounded in prior reasoning. Built on a Thinker–Talker architecture, Mini-Omni-Reasoner delivers fluent and precise spoken responses. On the Spoken-MQA benchmark, it achieves 19.1% improvement in arithmetic reasoning and 6.4% in contextual understanding, with zero latency. These results show that structured reasoning and natural spoken interaction can be effectively unified in a single framework.

## 1 INTRODUCTION

Reasoning is a fundamental faculty of human cognition, enabling precise, logically structured, and contextually grounded understanding of the external world (Simon, 1990). In natural communication and decision-making, humans frequently engage in internal deliberation prior to verbal expression, a strategy shown to enhance the factual accuracy, completeness, and reliability of responses. Inspired by this cognitive mechanism, recent advances in large language models (LLMs) (Jaech et al., 2024; He et al., 2025; Team, 2025; Guo et al., 2025) have formalized this strategy into the computational paradigm of *"thinking-before-speaking"*. In this formulation, models are prompted to construct an explicit and logically structured reasoning trace, which subsequently informs the final response. This reasoning-first formulation has demonstrated substantial benefits across a range of language tasks that demand structured explanation and logical consistency, such as mathematical reasoning.

While *"thinking-before-speaking"* paradigm has proven effective in textual domains, its direct extension to speech interfaces encounters inherent modality-specific constraints. Text affords *spatially parallel access*: readers can scan, skip, and selectively attend to different portions of content, enabling efficient comprehension of extended reasoning sequences at high reading speeds. In contrast, speech is consumed sequentially over time, constrained by the fixed-rate, streaming nature of auditory perception and human cognitive processing. Speaking out the reasoning trace before delivering an answer may burden listeners with verbose or low-utility content, delaying access to the core response. Conversely, keeping the reasoning silent leads to significant initial latency, as the model must complete its internal reasoning before producing responses, compromising interaction quality.

To bridge the gap between language reasoning and speech communication, where the conventional *"thinking-before-speaking"* paradigm proves ineffective for real-time spoken interaction, we introduce MINI-OMNI-REASONER, a novel speech reasoning framework founded on the principle of *"thinking-in-speaking"*. As shown in Figure 1, this formulation enables large speech-language models (LSLMs) to perform high-frequency internal reasoning in tandem with the real-time generation of semantically

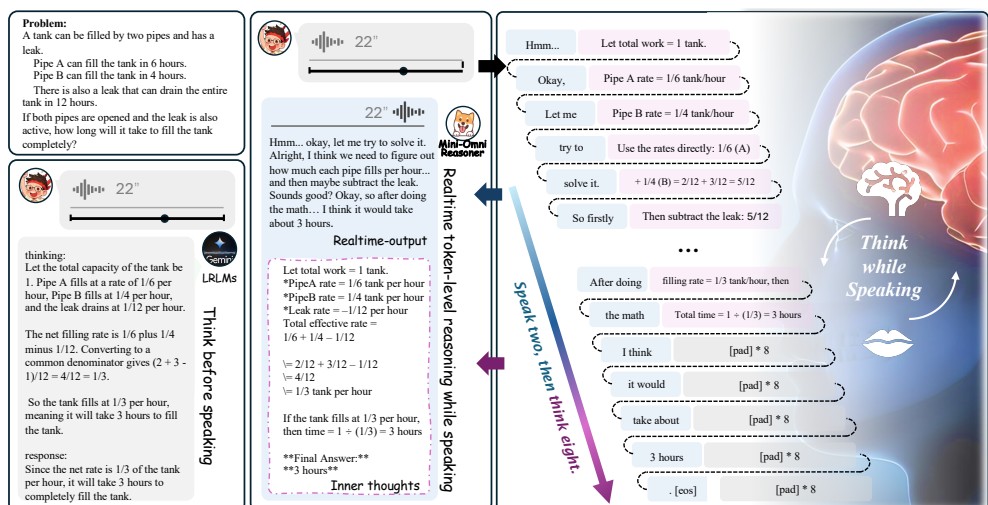

Figure 1: **Comparison of reasoning paradigms.** In the traditional *"thinking-before-speaking"* paradigm, models must complete reasoning before producing speech, leading to long latency or verbose spoken reasoning. In contrast, our *"thinking-in-speaking"* paradigm interleaves internal reasoning with continuous speech, enabling timely, fluent responses while preserving reasoning quality. This design leverages the gap between model inference throughput and audio playback speed to reduce latency and improve user experience without sacrificing depth.

informative spoken tokens. By decoupling the temporal resolution of internal inference from that of speech emission, our framework supports low-latency, cognitively aligned spoken interaction without sacrificing the depth, rigor, or interpretability of the underlying reasoning process.

MINI-OMNI-REASONER instantiates the *"thinking-in-speaking"* paradigm through an interleaved generation scheme that capitalizes on the discrepancy between model-side inference throughput and real-time audio playback constraints. Profiling results indicate that modern LSLMs can generate over 100 tokens per second on GPUs, while naturalistic audio playback typically requires only 12.5 tokens per second. To exploit this underutilized capacity, the model interleaves speech and reasoning tokens in a fixed proportion, enabling concurrent verbalization and latent inference. Specifically, we constrain the emission rate of spoken tokens to 20 per second for smooth playback and allocate the remaining generation bandwidth to reasoning. This yields a 2 *vs.* 8 speech-to-reasoning token ratio, derived directly from the inference budget rather than empirical heuristics. The system is built on the Thinker-Talker architecture (Xu et al., 2025a), ensuring that interleaved reasoning does not compromise the model's core language understanding or text-based reasoning performance.

To incentivize the reasoning capabilities of LSLMs under the *"thinking-in-speaking"* paradigm, we construct a data pipeline and introduce a large-scale dataset, SPOKEN-MATH-PROBLEMS-3M, tailored for audio-based mathematical reasoning. Building on prior evidence that mathematical tasks effectively elicit structured cognitive processes in language models, we curate an audio-based dataset of mathematical problems with difficulty comparable to the GSM8K (Cobbe et al., 2021) benchmark. A key challenge in this setting is *overshooting*, where the verbal output stream advances ahead of the internal reasoning process, leading to premature or hallucinated answers. To address this, we generate two temporally aligned streams for each problem: a fluent, human-readable output sequence and a symbolic, step-by-step reasoning trace. We introduce a prompting strategy that defers substantive content in the output stream while frontloading reasoning steps in the internal stream, thereby establishing a temporal buffer for inference. The resulting streams are tokenized, interleaved, and verified to ensure causal consistency, *i.e.*, no verbal content precedes its logical derivation. Upon this pipeline, we construct a dataset of 3 million audio-based mathematical reasoning samples by converting a broad collection of publicly available text-based datasets into speech format.

Given the immaturity of speech-language models and our aim to rigorously assess reasoning, we follow the LLM literature and validate our paradigm in the mathematically challenging domain. We conduct systematic experiments on six math word problem benchmarks, including Spoken-MQA (Wei et al., 2025), AddSub (Hosseini et al., 2014), SimpleEQ (Koncel-Kedziorski et al., 2015),

SimpleOP (Roy & Roth, 2016b), MultiArith (Roy & Roth, 2016a), and SVAMP (Patel et al., 2021b). On Spoken-MQA, MINI-OMNI-REASONER surpasses Qwen2.5-Omni-3B in arithmetic (64.9% → 77.25%, +12.4%) and reasoning (64.0% → 68.1%, +4.1%), while halving response length (42.9 vs. 116.1 words). Comparable or improved accuracy with notable efficiency gains is also observed across other datasets, with no overshooting after detokenization checks. These results highlight the effectiveness of our *"thinking-in-speaking"* paradigm, which interleaves reasoning and responses but only verbalizes the latter, thereby preserving correctness while enabling concise, real-time interaction.

## 2 INVOLVING REASONING IN SPOKEN DIALOGUE MODELS

We revisit the Thinker-Talker architecture, a state-of-the-art spoken dialogue paradigm. We then analyze how to incorporate reasoning into this architecture, illustrating the transition from the conventional *"thinking-before-speaking"* paradigm to our proposed *"thinking-in-speaking"* formulation.

### 2.1 THINKER-TALKER PIPELINE

The Thinker-Talker framework decouples audio understanding, linguistic inference, and speech synthesis. It consists of three core modules: an audio encoder, a Thinker LLM, and a Talker LLM. Given a raw audio input $\mathbf{x}_a$, the audio encoder first converts it into discrete audio tokens: $\mathbf{h}_{1:T}^a = \mathcal{E}_a(\mathbf{x}_a)$. These tokens, interpreted as linguistic actions, are passed to a Thinker LLM, which autoregressively generates a sequence of response tokens:

$$\mathbf{t}_{1:N}^{\text{resp}} = \mathcal{T}_{\text{thinker}}(\mathbf{h}_{1:T}^a) \tag{1}$$

Each generated response token $\mathbf{t}_j^{\text{resp}}$ is immediately mapped into audio tokens via the Talker LLM:

$$\mathbf{z}_j^a = \mathcal{T}_{\text{talker}}(\mathbf{t}_j^{\text{resp}}) \tag{2}$$

These audio tokens are concatenated to form a continuous stream: $\mathbf{z}_{1:J}^a = [\mathbf{z}_1^a; \mathbf{z}_2^a; \dots; \mathbf{z}_J^a]$.

To generate audible output, an audio decoder operates on fixed-size sliding windows over this stream. Each audio segment $\hat{\mathbf{x}}_i^a$ is reconstructed from a windowed slice of the audio token stream:

$$\hat{\mathbf{x}}_i^a = \mathcal{D}_a\left(\mathbf{z}_{s_i:s_i+\ell-1}^a\right) \tag{3}$$

where $s_i$ is the starting index of the $i$-th window and $\ell$ is the predefined audio token segment length. This streaming formulation enables real-time spoken interaction while maintaining modular separation between linguistic reasoning and audio synthesis. It also supports seamless integration of advanced reasoning capabilities within the Thinker module.

### 2.2 THINKING-BEFORE-SPEAKING

To explore reasoning integration, we start with the *"thinking-before-speaking"* paradigm. Here, the Thinker LLM is augmented to generate a latent reasoning sequence before emitting response tokens. Given the audio token sequence $\mathbf{h}_{1:T}^a$, the Thinker first generates: $\mathbf{t}_{1:M}^{\text{reason}} = \mathcal{T}_{\text{thinker}}(\mathbf{h}_{1:T}^a)$. Conditioned on both the audio and reasoning tokens, it then produces the verbal response: $\mathbf{t}_{1:N}^{\text{resp}} = \mathcal{T}_{\text{thinker}}(\mathbf{h}_{1:T}^a, \mathbf{t}_{1:M}^{\text{reason}})$. In this case, we consider two decoding strategies for the Talker LLM depending on how it handles reasoning tokens $\mathbf{t}_{1:M}^{\text{reason}}$ and response tokens $\mathbf{t}_{1:N}^{\text{resp}}$.

**Full Verbalization.** In this approach, both reasoning and response tokens are converted into audio:

$$\hat{\mathbf{x}}_{1:(M+N)}^a = \mathcal{D}_a(\mathcal{T}_{\text{talker}}([\mathbf{t}_{1:M}^{\text{reason}}; \mathbf{t}_{1:N}^{\text{resp}}])). \tag{4}$$

This produces a complete narration including reasoning and answer, but requires the listener to hear through reasoning content before the actual answer, introducing potential cognitive overload.

**Silent Reasoning.** Alternatively, the Talker LLM remains silent during reasoning token generation and only begins decoding when the first response token segment is available:

$$\hat{\mathbf{x}}_t = \begin{cases} \text{silent,} & \text{if } \mathbf{t}_i \in \mathbf{t}_{1:M}^{\text{reason}} \\ \mathcal{D}_{\text{audio}}\left(\mathcal{T}_{\text{talker}}(\mathbf{t}_i)\right), & \text{if } \mathbf{t}_i \in \mathbf{t}_{1:N}^{\text{resp}}, \end{cases} \tag{5}$$

This strategy ensures that only essential information is verbalized, improving clarity and efficiency, though it incurs a first-token delay due to the reasoning phase.

## 2.3 THINKING-IN-SPEAKING

To address the trade-off between reasoning depth and response latency, we introduce a novel *"thinking-in-speaking"* paradigm that interleaves reasoning and response generation. Unlike the conventional thinking-before-speaking approach, which delays response until reasoning is complete, our method enables the Thinker LLM to alternate between generating $p$ response tokens and $q$ reasoning tokens:

$$\mathbf{t}_{1:(p+q)\cdot K} = \bigcup_{i=1}^{K} \left\{ \mathbf{t}^{\text{resp}}_{(i-1)(p+q)+1}, \cdots, \mathbf{t}^{\text{resp}}_{(i-1)(p+q)+p}, \mathbf{t}^{\text{reason}}_{(i-1)(p+q)+p+1}, \cdots, \mathbf{t}^{\text{reason}}_{i(p+q)} \right\}. \quad (6)$$

During token prediction, the Talker LLM operates in a selective manner: it converts only the response segments into audio while remaining silent for the reasoning tokens. Compared to *thinking-before-speaking*, which waits for all $\mathbf{t}^{\text{reason}}$ to finish before any speech is produced, our interleaved generation scheme allows real-time response streaming while reasoning is still in progress.

Once a response token $\mathbf{t}^{\text{resp}}_i$ is generated, it passes the Talker for real-time conversion into speech:

$$\hat{\mathbf{x}}^{\text{a}}_i = \mathcal{D}_{\text{audio}} \left( \mathcal{T}_{\text{talker}}(\mathbf{t}^{\text{resp}}_i) \right). \quad (7)$$

This strategy exploits the empirical observation that token generation in autoregressive LLMs is significantly faster than real-time audio rendering. Thus, response segments can be emitted promptly while reasoning continues in the background, enabling continuous, low-latency interaction.

The $(p, q)$ ratio serves as a tunable parameter, balancing reasoning granularity with responsiveness, and can be adapted based on model throughput and deployment constraints. The full design and implementation of this *"thinking-in-speaking"* pipeline are elaborated in the following section.

## 3 MINI-OMNI-REASONER

Grounded in our *"thinking-in-speaking"* paradigm, we present Mini-Omni-Reasoner, a framework for real-time spoken dialogue with integrated reasoning. In this section, we detail three key aspects: *(1) Implementation*: the realization of thinking-in-speaking within audio–language models, including architectural design, token-ratio scheduling, and the use of special tokens; *(2) Data preparation*: an in-depth analysis of the training corpus together with the construction of a tailored synthesis pipeline; *(3) Training methodology*: a five-stage curriculum that progressively guides the base model toward end-to-end integration of reasoning and speech generation in the spoken modality.

### 3.1 IMPLEMENTATION DETAILS MINI-OMNI-REASONER

**Architecture.** As shown in Figure 2, Mini-Omni-Reasoner adopts a hierarchical Thinker–Talker architecture for real-time spoken reasoning. The core lies in the Thinker, which interleaves internal reasoning and response under the Thinking-in-Speaking paradigm. It comprises an audio encoder, an adapter, and a language model: raw audio is encoded into a semantic space and prefixed into the language model. The Thinker is initialized from Qwen2.5-Omni-3B, trained with an interleaved token-level objective, and then frozen to preserve reasoning capability. The Talker, a lightweight model of the same architecture, is trained from scratch with the SNAC tokenizer (Siuzdak et al., 2024) to transform response tokens into fluent, low-latency speech. When introducing the thinking-in-speaking mechanism on the Thinker, only response tokens are passed to the Talker as effective inputs. Thus, the speech synthesis process is identical to Mini-Omni (Xie & Wu, 2024), with no occurrence of overshooting between audio and text tokens. This modular separation of reasoning and speech generation enables Mini-Omni-Reasoner to unify logical inference with natural spoken output.

**Token-Level Thinking-in-Speaking.** We introduce how MINI-OMNI-REASONER implements token-level thinking-in-speaking within the Thinker module. Traditional LLMs (OpenAI, 2025; Guo et al., 2025; Xu et al., 2025a) typically follow a thinking-before-speaking strategy, where a full reasoning trace is generated before response. While effective in text, this paradigm causes excessive latency in speech generation, as the verbal response is delayed until reasoning completes. To address this, Mini-Omni-Reasoner adopts an interleaved generation strategy, where the model alternates between silent reasoning and spoken response tokens. We use a 2-to-8 ratio: the model emits 2 response tokens followed by 8 reasoning tokens in each cycle. This design enables continuous, fluent speech while preserving sufficient internal reasoning to support accurate response generation. This mechanism

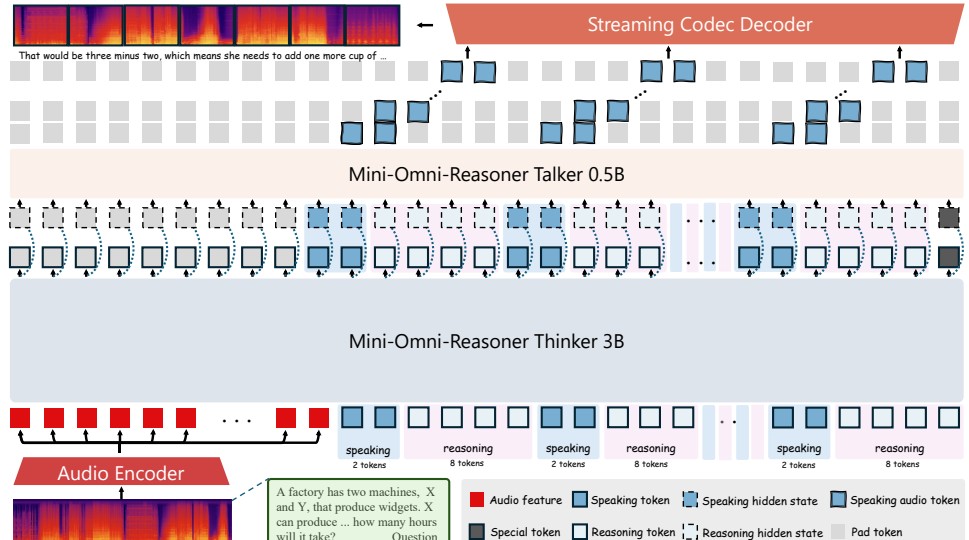

Figure 2: **Overview of Mini-Omni-Reasoner.** Given a raw audio instruction, the audio encoder converts it into language-space tokens, which pre-fill the Thinker LLM for autoregressive generation. The Thinker outputs an interleaved sequence of spoken response tokens and silent reasoning tokens. Response tokens are streamed to the Talker and decoded into speech in real time, while reasoning tokens remain unspoken but guide generation.

relies on two components. First, the interleaving ratio controls the trade-off between fluency and reasoning depth. Second, special control tokens are introduced to delineate reasoning and response segments during both training and inference, ensuring structured generation and stable alignment.

*Reasoning–Response Token Ratio Design.* The 2-to-8 ratio is chosen to balance latency, reasoning quality, and decoding controllability. First, short response blocks help avoid premature verbalization before reasoning is established. Second, the 2-to-8 schedule dedicates more capacity to internal inference, enhancing logical depth. Notably, this design still preserves a four-to-one ratio of reasoning tokens over output tokens, whereas dataset construction typically yields only a 1.5–2× ratio; this margin is critical to preventing *overshooting*, i.e., reasoning lagging behind speech output. Third, language models typically generates more than 100 tokens per second on a 3090 GPU, resulting in at least 20 spoken tokens per second, which is sufficient for smooth, real-time speech synthesis. Empirically, this configuration achieves a strong trade-off and is used as the default in Mini-Omni-Reasoner, with subsequent experiments further confirming its effectiveness in avoiding overshooting.

*Control Token Design.* To ensure stable alternation between reasoning and response, we investigate three control token strategies. **No explicit marker**: Training without any explicit token boundary fails, as the model drifts from the intended alternation. (textbfExplicit markers: Inserting emphasized split tokens destabilizes generation and causes misalignment. **Masked markers**: We insert split tokens that are masked from the loss computation during training. This proves most effective. During inference, we reintroduce split tokens to guide generation. Additionally, we append eight padding tokens after each reasoning block to stabilize Talker alignment and reinforce the 2-to-8 schedule.

## 3.2 SPOKEN-MATH-PROBLEM DATASET

A key prerequisite for enabling reasoning in speech is the construction of high-quality, temporally aligned training data. One central challenge is *anticipation drift*, where verbal responses appear before sufficient reasoning is completed, leading to semantic misalignment. To address this, we design a structured pipeline that tightly couples internal reasoning with coherent spoken outputs.

**Thinking-in-Speaking Formulation.** The *Thinking-in-Speaking* paradigm interleaves reasoning and response tokens to enable real-time generation. However, naive interleaving can lead to early response tokens appearing without sufficient prior reasoning. To enforce semantic precedence, we adopt a two-stage strategy. First, we introduce an *asynchronous alignment* scheme inspired by human dialogue patterns. Reasoning traces begin directly with logical content, while responses are encouraged to start with light contextual cues before transitioning to reasoning-supported content. This temporal

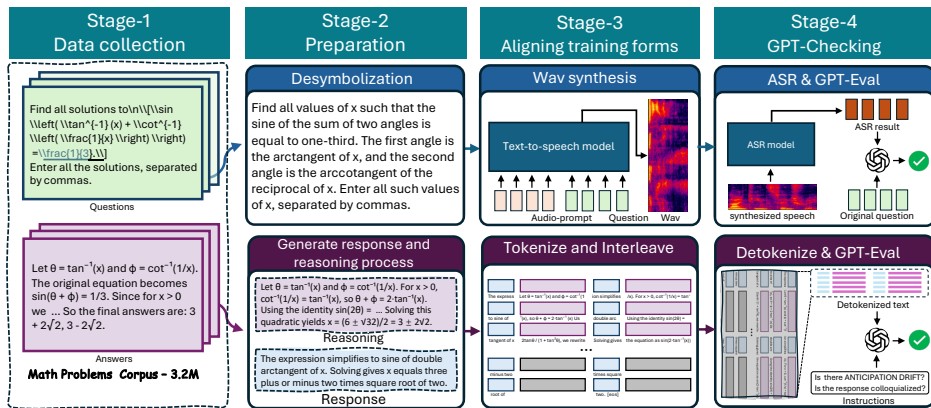

Figure 3: **Pipeline for Constructing the Spoken-Math-Problem-3M Dataset.** We aggregate large-scale math problems from public text datasets, reformulate them into spoken-style prompts, synthesize speech with TTS, and apply GPT-based verification to ensure fluency, coherence, and semantic fidelity.

offset ensures that response segments are semantically grounded in preceding reasoning. Second, we apply *sequence-level verification* to enforce local semantic coherence. Each reasoning–response pair is tokenized and interleaved using a 2 *vs*. 8 ratio, detokenized, and evaluated by a GPT-based checker. Only examples where reasoning correctly precedes response are retained.

**Data Construction Pipeline.** We build SPOKEN-MATH-PROBLEMS-3M, a 3M math word problems dataset, with the construction pipeline illustrated in Figure 3. Questions are resampled from high-quality text QA datasets, then rewritten into formal and spoken variants, with answers split into symbolic reasoning traces and concise responses. Reasoning is about twice the length of responses to preserve logical grounding. The rewritten questions are synthesized into audio with CosyVoice2-0.5B (Du et al., 2024). Reasoning and response segments are tokenized and interleaved at a fixed 2:8 ratio, preserving alternation. Finally, GPT-based verification ensures each response is logically supported, yielding a dataset that reflects the intended Thinking-in-Speaking paradigm. For more details, see the Appendix C.

## 3.3 TRAINING METHODOLOGY

Training Mini-Omni-Reasoner requires a staged pipeline to stabilize convergence and transfer reasoning from text to speech. We design five progressive stages (Figure 4). **Stage 1: Alignment.** Initialized from Qwen2.5-Omni-3B, we resolve architectural inconsistencies (*e.g.*, RoPE variants) and adapt interfaces. First, only the audio adapter is tuned on speech QA/dialogue while others are frozen; then all modules except the audio encoder are unfrozen to incorporate special tokens into the tokenizer. **Stage 2: Mixed Mathematical Pretraining.** The aligned model is enhanced for mathematical reasoning using "thinking-before-speaking" datasets in text and speech, ensuring strong reasoning before interleaved training. **Stage 3: Textual Thinking-in-Speaking.** Training begins in text, where the model alternates reasoning and response tokens. Only LM parameters are updated to internalize the interleaved structure. **Stage 4: Acoustic Thinking-in-Speaking.** Text queries are replaced with audio; the audio encoder is tuned while the LM remains fixed, transferring reasoning-augmented generation to speech. **Stage 5: Talker Training.** Finally, the frozen "thinker" is paired with a talker module trained to synthesize fluent speech from interleaved outputs, ensuring natural spoken responses while preserving logical grounding.

## 4 EXPERIMENTS

### 4.1 EXPERIMENTAL SETUP

**Training Setup.** Our training process build upon the mini-omni codebase, where we reconstruct the foundational model architecture from scratch. Specifically, we adopt the Qwen2.5-Omni encoder module as the audio encoder to extract speech features, and introduce a single linear adapter layer to bridge the audio encoder and the language model. The core language model is based on Qwen2.5-3B, which, together with the encoder and adapter, forms the MINI-OMNI-REASONER framework. To

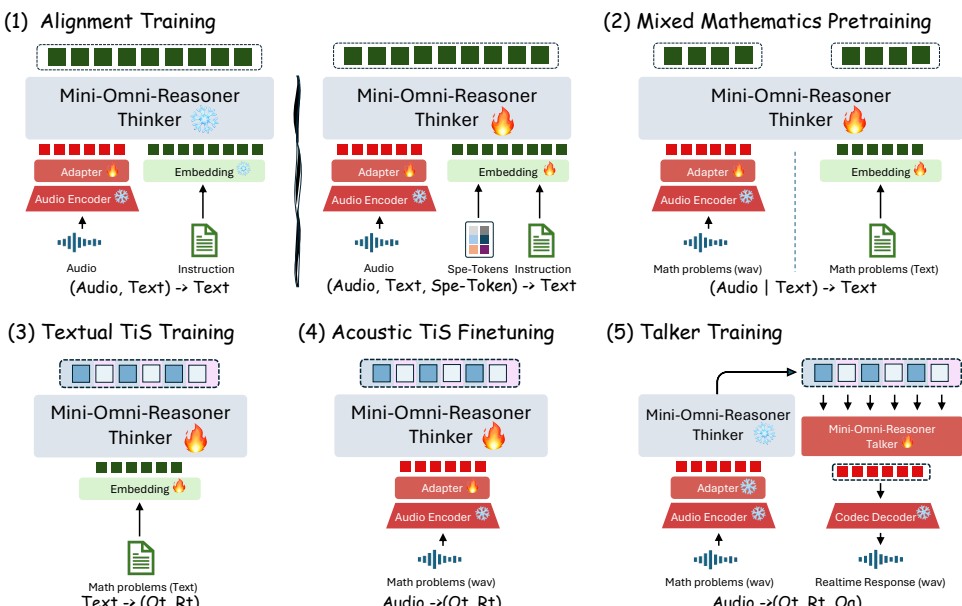

Figure 4: **Training Pipeline of MINI-OMNI-REASONER.** To enable interleaved reasoning and speaking, we progressively adapt the system through a multi-stage training process.

ensure parameter alignment and stable convergence, all model components are initialized from the corresponding modules of the pre-trained Qwen2.5-Omni-3B checkpoint. Training is conducted on 32 NVIDIA H100 GPUs, leveraging data parallelism for efficiency. We train on a large-scale dataset containing 3 million samples, running for 6 full epochs with a batch size of 64. The learning rate follows a cosine decay schedule, with the maximum learning rate set to 2e-4(details in Appendix B).

**Benchmark.** We evaluate our models on the Spoken-MQA (Wei et al., 2025) benchmark, which assesses spoken mathematical reasoning across two tasks: **Arithmetic** and **Reasoning**. The Arithmetic task focuses on basic numerical operations with minimal context, while the Reasoning task involves word problems requiring contextual interpretation and multi-step inference. It includes single-step samples from AddSub (Hosseini et al., 2014; Mishra et al., 2022) and SingleOp (Roy & Roth, 2016b), and multi-step samples from GSM8K (Cobbe et al., 2021) and SVAMP (Patel et al., 2021a), reflecting increasing complexity. Since Spoken-MQA

Table 1: Statistics of benchmarks with sub-task categories and sample counts.

| Benchmark | Sub-task | Difficulty | #Samples |
|---|---|---|---|
| Spoken-MQA | Arithmetic | Short | 118 |
| | Arithmetic | Long | 155 |
| | Reasoning | Single_step | 594 |
| | Reasoning | Multi_step | 1402 |
| Addsub | | | 395 |
| SingleEQ | Reasoning | Single_step | 508 |
| SingleOP | | | 562 |
| Svamp | | | 1000 |
| MultiArith | Reasoning | Multi_step | 600 |

covers only a subset of reasoning skills, we additionally evaluate our models on the full versions of AddSub, SingleEQ, SingleOp, SVAMP, and MultiArith. Sample statistics are provided in Table 1.

**Baselines.** We compare our model against three categories of baselines. **Cascade models** use Whisper-v3-large (Radford et al., 2023) for ASR followed by strong text-based LLMs such as Qwen2.5-Instruct-7B (Yang et al., 2024a) and Qwen2.5-Math-7B-Instruct (Yang et al., 2024b), serving as upper bounds with full text supervision. **Speech models** (Chen et al., 2024; Xie & Wu, 2024; Défossez et al., 2024; Fang et al., 2024; Wang et al., 2024; Chu et al., 2024; Xu et al., 2025a), with step-by-step prompting where applicable. Finally, we include our base model Qwen2.5-Omni-3B under both standard and "think step by step" decoding modes. This setup enables comparison across pipeline-based, end-to-end, and foundation-level models under a unified reasoning benchmark.

### 4.2 MAIN RESULTS

Table 2 reports results on Spoken-MQA. MINI-OMNI-REASONER achieves 92.9% on short-form and 66.1% on long-form arithmetic, surpassing cascade models (Radford et al., 2023) and conversational baselines (Xu et al., 2025a). Most speech models collapse on long-digit computation (e.g., Mini-Omni

Table 2: Spoken-MQA results (%). Best per column in bold. Models with * indicate that the prompt includes "please think step by step."

| Models | Size | Arithmetic | | | Reasoning | | | Avg |
|---|---|---|---|---|---|---|---|---|
| | | Short | Long | Avg | Single | Multi | Avg | |
| *Cascade* | | | | | | | | |
| Whisper-Qwen2.5-7B-Instruct | 7B | - | - | 70.0 | - | - | 72.5 | 72.2 |
| Whisper-Qwen2.5-Math-7B-Instruct | 7B | - | - | 77.3 | - | - | 86.7 | 85.6 |
| *Conversational Models* | | | | | | | | |
| SLAM-Omni | 0.5B | 0.0 | 0.0 | | 0.8 | 1.4 | 1.22 | 1.1 |
| Moshi | 7B | 0.0 | 0.0 | | 0.2 | 0.2 | 0.2 | 0.2 |
| LLaMA-Omni | 7B | 40.0 | 11.0 | 23.5 | 29.5 | 10.5 | 16.2 | 16.8 |
| Mini-Omni | 7B | 5.0 | 2.3 | 3.5 | 0.8 | 1.9 | 1.6 | 1.7 |
| Freeze-omni | 7B | 43.0 | 14.5 | 26.8 | 69.0 | 19.8 | 34.4 | 33.3 |
| GLM-4-Voice | 9B | 40.0 | 22.5 | 30.1 | 54.4 | 28.5 | 36.2 | 35.3 |
| Qwen2-Audio-7B-Instruct | 7B | 61.0 | 39.3 | 48.7 | 56.3 | 21.2 | 31.7 | 33.7 |
| Qwen2-Audio-7B-Instruct* | 7B | 43.0 | 31.2 | 36.3 | 55.4 | 22.5 | 32.3 | 32.7 |
| Qwen2.5-Omni-7B | 7B | 90.0 | 49.1 | 66.8 | 84.9 | 71.0 | 75.1 | **73.8** |
| Qwen2.5-Omni-7B* | 7B | 83.0 | 45.1 | 61.5 | 85.2 | **71.5** | **75.6** | 73.6 |
| *Baseline* | | | | | | | | |
| Qwen2.5-Omni-3B | 3B | 87.0 | 48.0 | 64.9 | 81.8 | 56.4 | 64.0 | 63.7 |
| Qwen2.5-Omni-3B* | 3B | 84.0 | 43.3 | 60.1 | 81.5 | 57.1 | 64.4 | 63.6 |
| *Ours* | | | | | | | | |
| Mini-Omni-Reasoner | 3B | **92.9** | **66.1** | **77.25** | **85.9** | 60.5 | 68.1 | 68.6 |

Table 3: Performance metrics on mathematical reasoning benchmarks. ACC: accuracy (%), Words: average token count, EFF: efficiency (ACC/Words).

| Benchmark | Mini-Omni-Reasoner | | | Qwen2.5-Omni-3B | | | Qwen2.5-Omni-3B(no think) | | |
|---|---|---|---|---|---|---|---|---|---|
| | ACC | Words | EFF | ACC | Words | EFF | ACC | Words | EFF |
| **Addsub** | 80.7 | 27.4 | 2.9 | 78.0 | **51.3** | **1.5** | **56.9** | 13.5 | **4.2** |
| | - | - | - | 3.5%↑ | 46.6%↓ | 93.3%↑ | 41.8%↑ | 103.0%↑ | 31.0%↓ |
| **SingleEQ** | 83.9 | 29.9 | 2.8 | 82.6 | 56.7 | **1.5** | 47.7 | **11.6** | 4.1 |
| | - | - | - | 1.6%↑ | 47.3%↓ | 86.7%↑ | 75.9%↑ | 157.8%↓ | 31.7%↓ |
| **SingleOP** | **90.1** | **26.6** | **3.4** | **88.2** | 74.1 | 1.2 | 19.0 | 12.4 | 1.5 |
| | - | - | - | 2.2%↑ | 64.1%↓ | 183.3%↑ | 379.5%↑ | 114.5%↑ | 126.7%↑ |
| **SVAMP** | 79.1 | 34.8 | 2.3 | 76.9 | 71.1 | 1.1 | 38.5 | 13.2 | 2.9 |
| | - | - | - | 2.9%↑ | 51.1%↓ | 109.1%↑ | 105.5%↑ | 163.6%↑ | 20.7%↓ |
| **MultiArith** | 85.4 | 31.1 | 2.7 | 86.8 | 71.4 | 1.2 | 46.7 | 15.2 | 3.1 |
| | - | - | - | 1.6%↓ | 56.4%↓ | 125.0%↑ | 82.9%↑ | 104.6%↑ | 12.9%↓ |

2.3%, LLaMA-Omni 11.0%). For reasoning, it obtains 85.9% on single-step and 60.5% on multi-step tasks, outperforming cascade and end-to-end open-source models, including Qwen2.5-Omni-7B. On average, it improves over the strongest cascade baseline by +4.0% in arithmetic and +4.1% in reasoning, despite its smaller size. Additional evaluations (Table 3) show slightly higher accuracy than the base model (83.8% vs. 82.5%) while halving response length (29.9 vs. 67.1 words). A no-reasoning variant of Qwen2.5-Omni-3B yields shorter outputs but severe accuracy drops (e.g., 88.2% → 19.0% on SVAMP), confirming the necessity of internal reasoning even without full verbalization. Case comparisons in Appendix A further illustrate output differences.

## 4.3 REASONING EFFICIENCY COMPARISON

Figure 5 compares reasoning accuracy and spoken length. MINI-OMNI-REASONER matches or outperforms larger models such as Qwen2.5-Omni-7B* on single-step (85.9%) and multi-step (60.5%) reasoning, while producing significantly shorter spoken responses (42.9 vs. 116.1 words). This efficiency stems from the *Thinking-in-Speaking* design: although total token generation is longer,

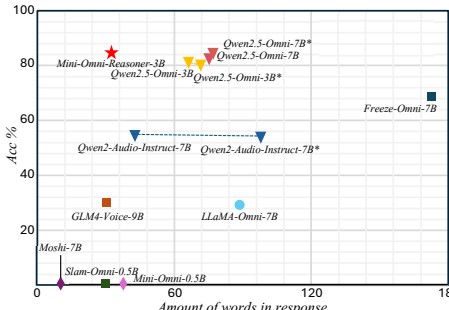 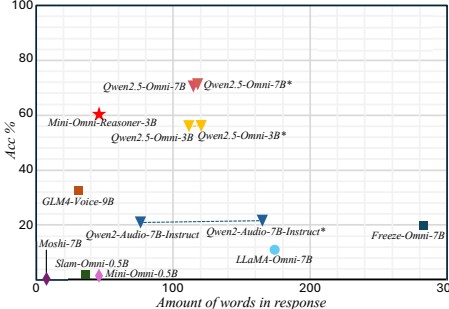

Figure 5: **Performance and response length on Spoken-MQA.** *Left:* single-step reasoning (simple); *right:* multi-step reasoning (hard). MINI-OMNI-REASONER-3B outperforms 3B-scale baselines and matches 7B models while generating shorter spoken responses.

only response tokens are verbalized, reducing audible content to around 25% of the full sequence. In contrast, models like Freeze-Omni (283.9 words) produce lengthy outputs without corresponding accuracy gains. Other baselines also show limited reasoning capability. Overall, MINI-OMNI-REASONER delivers a strong balance of accuracy and response efficiency.

Additionally, for the previously discussed *overshooting* phenomenon, we applied the same interleaving and detokenization checks to the test data on Spoken-MQA as done during data preparation. No cases showed reasoning lagging behind responses, and for complex problems reasoning length remained within about $1.5\times$ the response length. For more details, see the Appendix D.

## 5 RELATED WORK

**Speech LLMs.** Traditional speech systems rely on ASR–text–TTS pipelines, incurring high latency. Large audio LLMs (Zhang et al., 2023; Chu et al., 2024) address this by processing speech directly, while GPT-4o further enables ultra-low-latency interaction. MINI-OMNI (Xie & Wu, 2024) introduced a text-guided paradigm that generates speech tokens in parallel with text, inspiring follow-ups like FREEZE-OMNI, LLAMA-OMNI, and MOSHI. More recent systems, e.g., GLM4-VOICE (Zeng et al., 2024) and QWEN2.5-OMNI (Xu et al., 2025a), still externalize long reasoning chains as speech, causing latency, while our work internalizes reasoning as *inner thinking* to avoid unnecessary speech.

**Inference Scaling and CoT Reasoning.** CoT prompting (Wei et al., 2022) improves reasoning by step-by-step thinking. Building on this, models such as o1 (Jaech et al., 2024) and DeepSeek-R1 (Guo et al., 2025) extend reasoning through test-time scaling and RL optimization, inspiring methods like DAPO (Yu et al., 2025b) and GSPO (Yang et al., 2025). Recent approaches improve flexibility, robustness, and extend reasoning to multimodal settings (Gao et al., 2025; Hao et al., 2024; Xu et al., 2024; Huang et al., 2025; Xie et al., 2025).

**Reasoning Efficiency.** Long reasoning chains remain a bottleneck for real-time applications. In text LLMs, efficiency has been improved by length-controlled fine-tuning, shorter CoTs, token compression (Xia et al., 2025; Xu et al., 2025b), and RL-based adaptive reasoning (Fang et al., 2025; Dumitru et al., 2025). Hybrid models further decide when to reason or respond directly (Jiang et al., 2025; Yu et al., 2025a). However, these assume text outputs can be browsed asynchronously. In speech LLMs, long reasoning chains directly increase latency. This motivates our *"thinking-in-speaking"* paradigm, which internalizes reasoning without producing unnecessary speech.

## 6 CONLUSION

We presented Mini-Omni-Reasoner, a framework that integrates complex inner reasoning with real-time verbalization through a novel *"thinking-in-speaking"* paradigm. Unlike conventional thinking-before-speaking methods that incur high latency, our approach interleaves reasoning and response tokens, enabling fluent and timely speech while preserving logical consistency. To support this, we built the Spoken-Math-Problems-3M dataset and a progressive training pipeline that aligns reasoning with spoken output. Comprehensive experiments on Spoken-MQA show consistent improvements in arithmetic and contextual reasoning, with reduced response length and near-zero decoding latency. These results demonstrate that high-quality reasoning and natural spoken interaction can be achieved within a unified architecture, opening new directions for reasoning-aware speech systems.

## 7 ETHICS AND REPRODUCIBILITY STATEMENT

### 7.1 ETHICS STATEMENT

We follow the ICLR Code of Ethics.

**Subjects, Data, and Safety.** All speech data used for model training and evaluation has been rigorously screened. We do not use any unauthorized or improperly de-identified personal recordings or conversations. We strictly comply with data privacy regulations and relevant data licenses. We did not collect Personally Identifiable Information (PII) or biometric data (e.g., voiceprints without consent or proper de-identification). Any released media, such as speech excerpts or transcriptions used for qualitative analysis or demonstration, will be carefully checked to remove any identifying information (e.g., proper names, specific locations). All data sources we use (such as public academic speech benchmarks, crowdsourced datasets, or existing licensed data) will strictly adhere to their respective licensing terms.

**Fairness, Environment, and Disclosure.** Our models primarily process speech and language, and thus potential biases related to speaker attributes (e.g., accent, speaking rate, gender, age), socioeconomic background, or language/dialect may still appear. We are committed to responsible deployment of our models and will take steps to prevent their use for misinformation, deception, or privacy invasion. Upon acceptance, in line with the ICLR Code of Ethics, we will release our **code, checkpoints, data schemas**, and filtered datasets, along with licenses, clear documentation, and known limitations to facilitate reproduction.

### 7.2 REPRODUCIBILITY STATEMENT

We aim to maximize reproducibility. The main paper specifies model architectures with inference/decoding strategies (Section 3), training pipelines (Figure 4 and Section 4). The appendix details experimental setup, the data synthesis pipeline, and qualitative analysis (e.g., case studies). For datasets, we use open-access academic speech benchmarks as cited. For the datasets we collected or synthesized ourselves, we will release the **training datasets**, **model training scripts** (or detailed specifications), and **licensing information** upon acceptance.

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

# A  MODEL COMPARISON

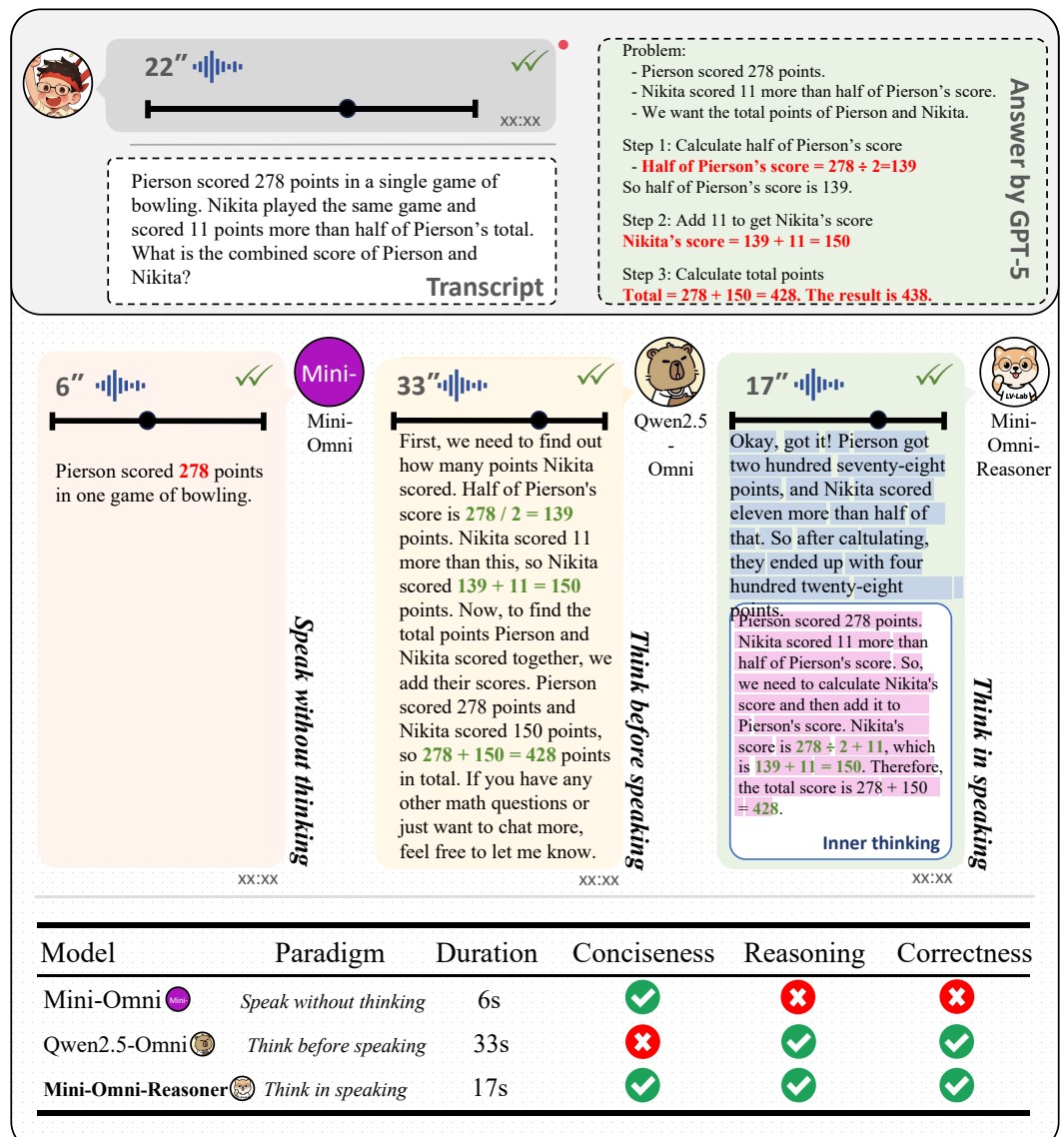

Figure 6: **Comparison of three speech model paradigms.** Early models like Mini-Omni perform simple dialogue with *speaking-without-thinking*. Qwen2.5-Omni, built on Thinker-Talker, supports reasoning but verbalizes the full chain, causing long and delayed outputs. Mini-Omni-Reasoner adopts *thinking-in-speaking*, delivering high-quality reasoning while keeping responses concise.

In this section, we provide a case study to compare the effectiveness of the proposed *"thinking-in-speaking"* paradigm against three alternative end-to-end speech models, as illustrated in Figure 6. Specifically, we consider: (i) Mini-Omni, which represents *"speaking-without-reasoning"* by directly mapping inputs to spoken answers without any reasoning traces, (ii) Qwen2.5-Omni-3B, which follows a *"thinking-before-speaking"* strategy by conducting full reasoning in the speech domain such that the entire reasoning trajectory is synthesized into speech, and (iii) MINI-OMNI-REASONER, our model, which adopts the *"thinking-in-speaking"* paradigm by interleaving reasoning tokens and response tokens, while only synthesizing the response into speech. The results reveal clear differences across paradigms. Models like Mini-Omni, despite achieving highly efficient responses,

consistently fail to ensure correctness due to the absence of reasoning. In contrast, Qwen2.5-Omni-3B successfully produces accurate answers by synthesizing its complete reasoning process, but this leads to extremely long spoken outputs, requiring tens of seconds for users to obtain the final answer. MINI-OMNI-REASONER achieves a favorable balance: although it generates more reasoning tokens than Mini-Omni, it drastically reduces response latency by using concise phrases (e.g., "after calculating") to summarize the computation, thereby halving the overall response time while preserving correctness. Finally, we summarize the comparison in the table at the bottom, which demonstrates that *"thinking-in-speaking"* combines the correctness of reasoning-based paradigms with the efficiency of direct-answering approaches.

## B  TRAINING ANALYSIS

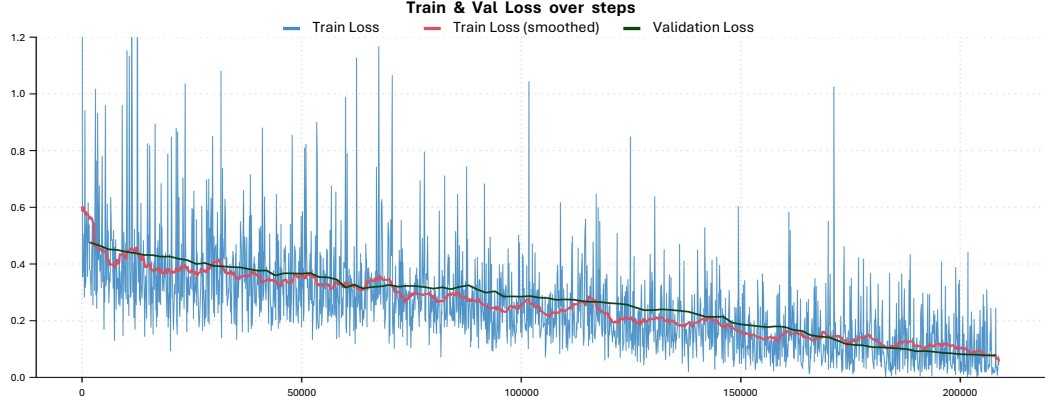

Figure 7: Training and Validation Loss Curves.

We initially hypothesized that frequent alternation between the "speak" and "think" modes might destabilize training by perturbing the data distribution and impeding convergence. Nevertheless, the loss curves presented in Figure 7 indicate stable optimization. The training loss decreases from approximately 0.6 to 0.1 over 200k steps, with smoothed curves confirming a consistent downward trend. The validation loss exhibits a nearly identical trajectory, converging closely with the training loss in later stages. This steady and coherent reduction, without signs of divergence or instability, demonstrates that MINI-OMNI-REASONER successfully mitigates the potential challenges associated with alternating "speak" and "think" modes, thereby validating the robustness of our training methodology.

## C  DETAILS OF THE DATA GENERATION PIPELINE

### C.1  TEXT TO SPEECH PIPELINE

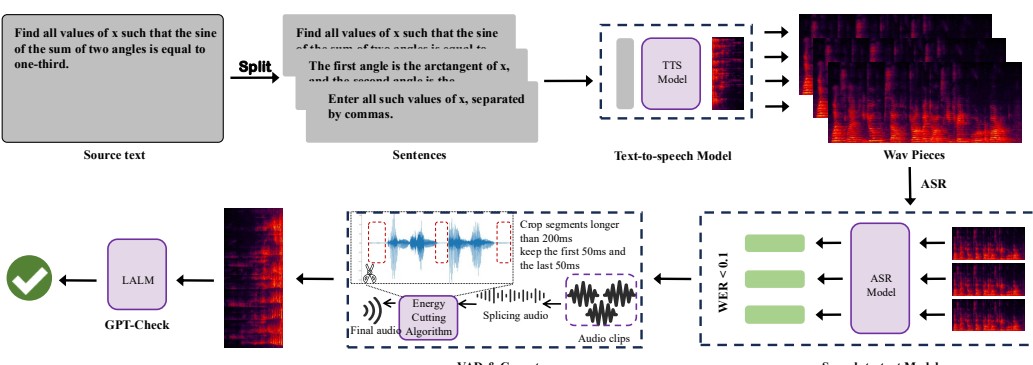

Figure 8: TTS pipline.

In this section, we detail the text-to-speech (TTS) pipeline employed for the construction of the **Spoken-Math-Problem-3M** dataset. Extensive preliminary experiments revealed that prevailing TTS models exhibit suboptimal performance when tasked with synthesizing long, intricate mathematical problems containing multiple sentences. Specifically, these models frequently introduce undesirable artifacts such as prolonged inter-sentence pauses, a decline in synthesis accuracy, and a lack of phonetic clarity. To mitigate these deficiencies, we devised a multi-stage TTS synthesis pipeline, which systematically integrates five discrete processes: sentence splitting, single-sentence TTS synthesis, ASR-based filtering, energy-based trimming, and a final quality assessment facilitated by a large language model.

*Sentence Splitting*  The initial stage involves segmenting lengthy sentences at each punctuation mark. Following each split, a verification is performed to ensure that all resulting sub-sentences contain fewer than ten words. Should any sub-sentence exceed this predefined length, the original sentence is re-segmented until the criterion is satisfied. Our empirical findings indicate that an overly granular splitting strategy detrimentally impacts synthesis efficiency, while an overly coarse approach yields speech of inferior quality. The proposed methodology strikes a critical balance between these two trade-offs.

*Single-Sentence TTS Synthesis*  For the synthesis of individual sentences, we utilize the CosyVoice-0.5B model (Du et al., 2024). We observed that decomposing long, complex linguistic units into discrete sentences effectively eliminates issues commonly associated with multi-sentence synthesis, such as erroneous pauses and a significant degradation in accuracy at sentence terminals. This approach is instrumental in ensuring the high fidelity and reliability of our synthesized dataset.

*ASR-Based Filtering*  While an initial evaluation considered Whisper (Radford et al., 2023) for the ASR filtering stage, its accuracy on longer audio segments was found to be notably low. Consequently, we adopted SenseVoice (An et al., 2024) as our ASR model of choice. We compute the Word Error Rate (WER) between the ASR-transcribed text and the ground-truth text. A sub-sentence is deemed successfully synthesized if its WER is below 0.1. Any audio segment that fails this validation is re-routed to the single-sentence TTS synthesis stage for reprocessing until it meets the filtering criteria.

*Energy-Based Trimming*  Upon successful synthesis of all individual audio segments, they are concatenated sequentially. An energy-based analysis is subsequently performed on the complete

audio file. Regions where the energy level falls below a predetermined threshold are trimmed, as these typically correspond to silent gaps introduced during synthesis that can disrupt the natural prosodic flow.

*Final GPT-Based Assessment* The energy-trimmed audio is then provided as a prompt to a large language model (GPT). The GPT model performs a final qualitative assessment to confirm that the overall quality of the synthesized speech is acceptable and meets the dataset's requirements.

## C.2   PROMPT FOR GENERATING REASONING DATA

> We are generating data that simulates a human thinking-and-speaking process while solving math problems. You will be given a math problem along with its standard solution.
>
> Your task is as follows:
>
> First, generate the most efficient yet logically sound reasoning process to solve the problem.
>
> Then, produce a spoken-style verbal response that expresses the full thought process and answer in a natural, conversational way.
>
> Most importantly, keep the following points in mind:
>
> 1.The reasoning process should be 2 to 3 times longer than the spoken response.
>
> 2.In natural speech, we tend to "think" 2–3 times more words than we actually say—so do not overgenerate. In other words, your spoken output must not exceed roughly twice the amount of content derived from your internal reasoning.
>
> My suggestion is: during the reasoning phase, focus on solving directly and efficiently; during the spoken phase, you may choose to expand a little—if the reasoning is short (e.g. for simple problems), give a quick explanation before the final answer; if the reasoning is long, you can describe the problem, outline your thinking strategy, and go through early steps to delay reaching the answer.
>
> You can think of it this way: in the thought part, 'put reasoning first—avoid unnecessary chatter and quickly work toward the answer'; in the output part, 'delay the reasoning'—start with some descriptive content or even early-stage thoughts from the thought section to buy time before reaching the final answer.
>
> Format: `<thought>`...`</thought><output>`...`</output>`.
>
> Here is a sample:
> ```
> <thought>With 5 people, each handshake is a unique pair.
> The number of such pairs is C(5, 2) = 5×4/2 = 10.  Or count
> manually:  first person shakes 4 times, second 3 (excluding
> previous), and so on:  4+3+2+1 = 10.</thought>
> <output>There are five people, and each handshake is between
> a unique pair.  So it's just five choose two, which is ten
> handshakes in total.</output>.
> ```
>
> Notice: the output content will be used to generate speech, so use only English words, with no digits, symbols, or even hyphens—every part must be readable words.
>
> Here is the question:
> **\*\*\* question here \*\*\***.
>
> Here is the answer process you can refer to:
> **\*\*\* answer here \*\*\***.
>
> Begin with `<thought>` and follow strictly with my format and sample. Give me a string not json format.
>
> Keep the think process shorter than 4-5 sentences, make it don't be too long and super efficient, and make it directly begin with math reasoning and do not say something like 'the problem is about... I will ...' these are useless in thinking process.
>
> Let the `<thought>` part be 1-2 times longer than the `<output>` part, and in the `<thought>` section, you can use all mathematical symbols.

### C.3 PROMPT FOR COLLOQUIAL MATH PROBLEMS

> You are an AI assistant whose job is to the symbols from a math problem and turn it into a problem that can be read out.
>
> Be careful not to change the original sentence structure, try to maintain the original structure.
>
> Follow these guidelines:
> 1.Describe every quantity in words. For example, say "four T-shirts" instead of "$4\times$ T-shirts," and "one hundred fifty dollars" instead of "$150."
>
> 2.Spell out operations and relationships. Use phrases like "multiplied by," "added to," or "in total" rather than symbols like "$\times$," "$+$," or "$=$."
>
> 3.Keep units and discounts explicit. Always include words like "dollars," "percent," "pairs," or "each."
>
> Here is the original math problem you can refer to:
> **\*\*\* math problem here \*\*\***.

### C.4 PROMPT FOR SCREENING DATA VALIDITY

> The following are two texts. 'user' is the original math problem, and 'text' is obtained after text-to-speech synthesis and speech recognition.
>
> Determine whether 'text' fully includes all key information from 'user', especially math conditions, numbers, and units.
>
> If yes, reply with true; otherwise, reply with false.
>
> Be lenient, but do not allow missing conditions or misread problems. There may be a lot of differences between the two texts because of different text-to-speech synthesis and speech recognition techniques. Focus on the information.
>
> Here is the original text you can refer to:
> **\*\*\* original text here \*\*\***.
>
> Here is the generated text you can refer to:
> **\*\*\* generated text here \*\*\***.

# D CASE STUDY

Table 4: Example of Mini-Omni-Reasoner's response of the most difficult data in the test set.

| Question | Corrected Response | Corrected Inner Thinking |
|---|---|---|
| Vincent can buy flowers in packages of 3 for $2.50 or in packages of 2 for $1. How much money does he save by buying 18 flowers at the better price? | Good question! Let's quickly sort out the cost for buying 18 flowers in both ways first. If Vincent picks packages of 3 (each $2.50), that'll cost $15 total, and if he goes for packages of 2 (each $1), it'll only be $9. Obviously, packages of 2 are the better deal here. After figuring out both costs, it's easy to see he saves $6 by choosing that option. | First, I need to figure out the two possible ways Vincent can buy 18 flowers, calculate the total cost for each way, compare the two costs to find the better price, and then compute the savings. For the first way (packages of 3), since each package has 3 flowers, the number of packages needed is 18 ÷ 3 = 6, and with each package costing $2.50, the total cost is 6 × $2.50 = $15. For the second way (packages of 2), each package has 2 flowers, so the number of packages needed is 18 ÷ 2 = 9, and with each package costing $1, the total cost is 9 × $1 = $9. Then, comparing $15 and $9, $9 is cheaper, so the better price is the package of 2, and the savings is the difference between the two costs, which is $15 - $9 = $6. |
| Out of the 200 Grade 5 students, 2/5 are boys and 2/3 of the girls are in the girl scout. How many girls are not in the girl scout? | Good question! Let's start by figuring out the key numbers here. First, two fifths of the two hundred Grade Five students are boys, so that's eighty boys total. Subtracting that from two hundred means there are one hundred and twenty girls. Then, since two thirds of these girls are in the girl scout, that's eighty girls in the scout. After working through those numbers, it's clear that the number of girls not in the girl scout is forty. | First, I need to find the number of boys first because the question is about girls, so subtracting boys from total students gives the number of girls. The number of boys is 2/5 of 200, which is calculated as 200 × (2/5) = 80. Then, the number of girls is total students minus boys: 200 - 80 = 120. Next, since 2/3 of the girls are in the girl scout, I need to find that number first to get the girls who are not in it. The number of girls in the girl scout is 120 × (2/3) = 80. Finally, the number of girls not in the girl scout is total girls minus girls in the scout: 120 - 80 = 40. |
| Gary likes to walk around the edge of the local park, which is a rectangle that measures 1.5 miles by 6 miles. If he walks at 3 miles/hour, how many hours does he spend walking? | Good question! Let's work through this step by step. First, since Gary walks around the edge of the rectangular park, we need to find the park's perimeter first. Then, we just divide that perimeter by Gary's walking speed, and that will give us the time he spends walking. After calculating, we'll find that the time he spends is five hours. | The park is a rectangle, so to find the distance Gary walks (which is the perimeter of the rectangle), we use the perimeter formula for a rectangle: 2 times length plus 2 times width. Here, the length is 6 miles and the width is 1.5 miles, so that's 2 times 6 plus 2 times 1.5. Calculating that gives 12 plus 3, which equals 15 miles. Gary walks at a speed of 3 miles per hour, so to find the time spent walking, we divide the total distance (perimeter) by his speed: 15 miles divided by 3 miles per hour, which equals 5 hours. |

| Question | Corrected Response | Corrected Inner Thinking |
|---|---|---|
| Steve decides to start eating more tomatoes and decides to grows his own cherry tomatoes. He eats twice as much as his girlfriend. He eats 6 per day. If a vine can produce 3 tomatoes per week how many vines does he need? | Good question! Let's break this down step by step. First, we need to find out how many tomatoes Steve's girlfriend eats each day, since Steve eats twice as much as her and we know Steve eats six per day. Once we have her daily amount, we can find the total number of tomatoes Steve and his girlfriend eat together each day. Next, we'll calculate their total weekly tomato consumption. Finally, we'll divide that weekly total by how many tomatoes each vine produces per week to find out how many vines Steve needs. After working through these steps, we'll see that Steve needs twenty-one vines. | First, find how many tomatoes Steve's girlfriend eats per day: Steve eats twice as much as her, and Steve eats 6 per day, so his girlfriend eats $6 \div 2 = 3$ tomatoes per day. Then, calculate their total daily tomato consumption: Steve's 6 plus his girlfriend's 3 equals 9 tomatoes per day. Next, find their weekly consumption (since a week has 7 days): 9 tomatoes/day $\times$ 7 days = 63 tomatoes per week. Each vine produces 3 tomatoes per week, so the number of vines needed is total weekly tomatoes divided by tomatoes per vine: $63 \div 3 = 21$ vines. |
| There are 36 penguins sunbathing in the snow. One-third of them jump in and swim in the ocean. Another one-third go inside the cave to eat their dinner. How many penguins are still left sunbathing? | Good question! Let's work through this step by step. First, we know some penguins leave to swim and others to eat dinner—both groups are one-third of the total sunbathing penguins. So first, we can figure out how many penguins leave in total, then subtract that from the original number to find how many stay. After calculating, we'll see that the number of penguins still left sunbathing is twelve. | First, find the number of penguins that jump into the ocean: that's one-third of 36, so 36 divided by 3 equals 12 penguins. Then, find the number of penguins that go to eat dinner: that's also one-third of 36, so another 36 divided by 3 equals 12 penguins. Next, calculate the total number of penguins that leave sunbathing: 12 (swimming) plus 12 (eating) equals 24 penguins. Finally, subtract the number of penguins that left from the original number of sunbathing penguins: 36 minus 24 equals 12 penguins. So 12 penguins are still left sunbathing. |
| A car in the fast lane is traveling at 60 miles/hour. A car in the slow lane is traveling at half that speed. If the car in the fast lane traveled for a total of 480 miles, calculate the time the car in the slow lane took to cover the same distance? | Good question! Let's walk through this together. First, we need to find out how fast the car in the slow lane is going, since we know it's half the speed of the fast lane car. Once we have the slow lane car's speed, we can use the distance it needs to cover (which is the same as the fast lane car's distance) to find the time it takes. After working through these steps, we'll see the time the slow lane car took is sixteen hours. | First, calculate the speed of the car in the slow lane: it's half the speed of the fast lane car, which is 60 miles/hour. So that's 60 divided by 2, which equals 30 miles/hour. Next, we know both cars cover the same distance of 480 miles, and time is calculated by dividing distance by speed. So for the slow lane car, time = distance / speed = 480 miles / 30 miles/hour. Calculating that gives 16 hours. So the car in the slow lane took 16 hours to cover 480 miles. |

# E    THE USE OF LARGE LANGUAGE MODELS (LLMS)

We used Large Language Models (LLMs) to support several aspects of this work. This included using LLMs to assist with data generation , performing data screening and quality filtering, and for validation checks on generated outputs. Additionally, an LLM provided final-stage assistance in refining the manuscript's grammar and style. We explicitly state that all core model development and scientific conclusions were the result of our human research team's independent efforts, and we maintained stringent final oversight and validation for all LLM-assisted tasks.

