# OpenReview forum: "Mini-Omni-Reasoner: Token-Level Thinking-in-Speaking in Large Speech Models"
_ICLR.cc/2026/Conference — ICLR 2026 Conference Withdrawn Submission_

### Official Review · Reviewer_7FHk · 2025-10-30

**Soundness:** 3
**Presentation:** 3
**Contribution:** 3
**Rating:** 6
**Confidence:** 3

**Summary:**

This paper introduces Mini-Omni-Reasoner, a large speech-language model designed under a new “thinking-in-speaking” paradigm. Instead of the conventional thinking-before-speaking strategy that delays spoken responses until reasoning completes, Mini-Omni-Reasoner interleaves reasoning and response tokens during generation. This enables real-time reasoning and fluent spoken responses with near-zero latency. The authors also release a large-scale dataset, Spoken-Math-Problems-3M, and design a 5-stage training pipeline to align reasoning with spoken output. On spoken mathematical reasoning benchmarks (e.g., Spoken-MQA, AddSub, SVAMP), the model significantly outperforms prior LSLMs and performs comparably to or better than larger text-speech models, achieving both shorter response length and improved reasoning accuracy.

**Strengths:**

1. The “thinking-in-speaking” formulation is clearly motivated by modality constraints and addresses the latency issue in the “thinking-before-speaking” paradigm. This is an interesting conceptual shift with potential implications beyond math tasks.
2. The paper combines algorithmic design (token-level interleaving), dataset construction (Spoken-Math-Problems-3M), and a new model (Mini-Omni-Reasoner). Together they demonstrate an end-to-end system for real-time reasoning in speech.
3. The paper is well-written and easy to follow.

**Weaknesses:**

1.Although the 2:8 speech-to-reasoning token ratio is justified by hardware throughput, hardcoding this ratio limits adaptability. If deployment hardware or inference speed changes, both model and dataset construction would need to be redone. It would strengthen the work to include an ablation or adaptive mechanism for this ratio. In related LLM work such as Quiet-STaR [1], increasing the number of “thinking” tokens can systematically improve reasoning. I am wondering if this is case for Mini-Omni-Reasoner.
2. The main evaluation is restricted to spoken math reasoning. While this is a clean domain for analysis, it is unclear whether “thinking-in-speaking” generalizes to other reasoning-heavy but natural conversational tasks. The claimed fluency and precision improvements are qualitatively supported, but more quantitative user or perceptual studies would be valuable.
3. The paper claims that detokenization checks guarantee “no overshooting,” but the guarantee seems heuristic—based on alignment checks and empirical verification rather than a theoretical bound.

[1] Quiet-STaR: Language Models Can Teach Themselves to Think Before Speaking, Zelikman et al., COLM 2024

**Questions:**

1. How can the data construction pipeline ensure there is truly no overshooting issue? The paper mentions "We introduce a prompting strategy that defers substantive content in the output stream while frontloading reasoning steps in the internal stream", but this seems heuristic rather than provable.

---

> ### Author Response · Authors · 2025-11-22
> **Rebuttal by Authors (Part-1)**
>
> Dear Reviewer 7FHk:
>
> Thank you for your valuable feedback and recognition of our work on Mini-Omni-Reasoner! We will address each of your questions in detail below.
>
> > Weakness 1: Although the 2:8 speech-to-reasoning token ratio is justified by hardware throughput, hardcoding this ratio limits adaptability. If deployment hardware or inference speed changes, both model and dataset construction would need to be redone. It would strengthen the work to include an ablation or adaptive mechanism for this ratio. In related LLM work such as Quiet-STaR [1], increasing the number of “thinking” tokens can systematically improve reasoning. I am wondering if this is case for Mini-Omni-Reasoner.
>
> Thank you for this insightful comment and for pointing us to related work such as Quiet-STaR. We agree that the relationship between reasoning token allocation and performance, as well as the possibility of adaptive mechanisms, represents an important direction for future research.
>
> Our work focuses on enabling real-time internal reasoning in spoken interaction, where latency constraints are fundamentally different from those in text-only LLM settings. The **2:8 speech-to-reasoning token ratio** was not arbitrarily chosen. It reflects a balance derived from hardware throughput limits and real-time deployment requirements. **Our measurements show that on RTX 3090, L40s, and RTX 4090 GPUs, the generation throughput reaches 30.4, 37.8, and 56.5 tokens per second respectively.** With the 2:8 ratio, the system generates approximately **11.3 output tokens and 45.2 reasoning tokens per second**, which maintains stable real-time interaction while operating near the upper bound of achievable throughput. Ratios such as 1:9 require throughput levels that only H100-class hardware can reliably support, making deployment on widely accessible consumer hardware impractical.
>
> To address the reviewer’s concern regarding adaptability, **we conducted additional experiments beyond the original 2:8 configuration. We evaluated four alternative ratios: 3:7, 1:9, 2:12, and 2:16, and compared them on the Spoken-MQA benchmark** and the result is illustrated in Table 1.
>
> Table 1: Performance Comparison Across Different Speech-to-Reasoning Token Ratios on **Spoken-MQA**
>
> | Token ratio | 3:7  | 2:8(ours)  | 1:9  | 2:12 | 2:16 |
> |-------------|------|------|------|------|------|
> | Performance | 64.2 | 68.6 | 67.4 | 69.1 | 69.3 |
>
>
> The results reveal a clear trend:
> - When the ratio becomes more restrictive on reasoning, such as 3:7, performance drops noticeably, indicating that insufficient reasoning space undermines coherent chain-of-thought generation.
> - **Increasing the reasoning proportion beyond 2:8 yields only marginal improvements**. The progression from 1:9 to 2:12 and 2:16 produces small gains rather than substantial advances, suggesting that **2:8 already provides sufficient reasoning capacity for the current task difficulty and model scale**.
>
> From a deployment perspective, larger reasoning allocations substantially affect real-time generation. On RTX 4090, the 2:12 and 2:16 settings approach or reach the throughput limit for continuous audio output, and 1:9 exceeds what consumer GPUs can sustain. **Considering both performance and practical deployment, we believe that 2:8 offers a robust and accessible trade-off for real-world systems.**
>
> We acknowledge that these findings are conditioned by the difficulty of current benchmarks. Most voice-based reasoning tasks, including those comparable to GSM8K, do not yet require extensive reasoning depth for strong performance. As voice models evolve and more challenging reasoning tasks emerge for large speech-language models, we expect the optimal allocation to shift. Exploring adaptive mechanisms that adjust the ratio based on task complexity or hardware capabilities is an exciting direction that we plan to pursue.
>
> We sincerely appreciate the reviewer’s suggestion, which motivated a more comprehensive analysis of this hyperparameter and strengthened the contribution of the work.

---

> ### Author Response · Authors · 2025-11-22
> **Rebuttal by Authors (Part-2)**
>
> > Weakness 2: The main evaluation is restricted to spoken math reasoning. While this is a clean domain for analysis, it is unclear whether “thinking-in-speaking” generalizes to other reasoning-heavy but natural conversational tasks. The claimed fluency and precision improvements are qualitatively supported, but more quantitative user or perceptual studies would be valuable.
>
> Thank you for this thoughtful suggestion. We fully agree that spoken math reasoning represents a clean and controlled domain, and that broader evaluation is important for assessing the generality of the proposed paradigm. The primary goal of this work is to investigate real-time reasoning in conversational settings, a direction that remains underexplored in the community. We therefore began with mathematical reasoning, where the reasoning chain is most explicit and the effects of Thinking-in-Speaking can be clearly isolated.
>
> **In response to the reviewer’s concern, we have extended our evaluation beyond mathematics to more natural and diverse spoken dialogue tasks.** To support this, **we additionally curated approximately 800,000 high-quality text-based QA pairs for fine-tuning.** The overall training pipeline remains consistent, with two modifications: daily conversational QA does not require deep reasoning, so long reasoning generation is not enforced; and the data were strictly filtered to remove instances that contain privileged or oracle-style reasoning. **The model was trained for two epochs with a maximum learning rate of 2e-4.**
>
> We then followed the evaluation protocols of Vita-Audio [2] and Qwen2.5-Omni [3], conducting systematic experiments on Llama-Questions, Web-Questions, AlpacaEval, CommonEval, SD-QA, and IFEval [4].
>
> Table2: Performance of Various Models on Knowledge QA and Fact Reasoning & Instruction Following Tasks.
> | Model               | Mini-Omni-Reasoner | Qwen2.5-Omni(3B) | GLM-Voice | Vita-Audio-Boost |
> |---------------------|--------------------|------------------|-----------|------------------|
> | **Knowledge QA**    |                    |                  |           |                  |
> | Llama-question      | 73.2               | **74.3**             | 64.7      | 68.7             |
> | web-questions       | **71.0**               | 68.7             | 55.0      | 32.9             |
> | **Fact Reasoning & Instruction Following** |                    |                  |           |                  |
> | AlpacaEval          | **4.46**               | 4.32             | 3.97      | 3.56             |
> | CommonEval          | 3.98               | **4.00**             | 3.42      | 3.24             |
> | IFEval              | **61.9**               | 53.5             | 25.92     | 49.9             |
> | SD-QA               | **59.6**               | 56.4             | 36.98     | 42.2             |
>
>
> **The results show that the model maintains competitive performance on knowledge-oriented QA benchmarks, achieving 73.2 on Llama-Questions and 71.0 on Web-Questions, comparable to the base model Qwen2.5-Omni-3B.** This consistency indicates that **introducing explicit reasoning does not harm conversational quality or factual reliability.**
>
> More importantly, the benefits of Thinking-in-Speaking become more pronounced in tasks requiring logical consistency and precise instruction adherence:
> - On AlpacaEval, the model reaches **4.46**, outperforming Qwen2.5-Omni-3B, GLM-Voice [5], and Vita-Audio-Boost.
> - On IFEval, which emphasizes complex instruction following, the model achieves **61.9**, substantially higher than all baselines, reflecting improved reasoning stability.
> - On CommonEval, the model achieves **4.00**, slightly higher than Qwen2.5-Omni-3B and clearly better than speech-focused baselines, demonstrating that the paradigm preserves general factual reasoning.
> - On SD-QA, which requires grounding in complex audio contexts, the model achieves **59.6** and exceeds all compared systems, showing stronger anchoring and coherence under challenging multimodal conditions.
>
> **We also conducted a user study to assess perceived reasoning quality in voice-based interactions.** Ten questions with correct answers from both models were presented as audio to thirty users through a blinded online interface. **Twenty-five preferred Mini-Omni-Reasoner** for its more natural and concise reasoning, **three favored Qwen2.5-Omni**, and **two were neutral**, suggesting that users tend to prefer concise, coherent spoken reasoning over fully detailed explanations.
>
> **These results provide quantitative evidence that Thinking-in-Speaking generalizes beyond structured mathematical reasoning and offers consistent gains in instruction fidelity, reasoning stability, and factual grounding across a variety of natural conversational tasks.** We appreciate the reviewer’s suggestion and believe that these expanded evaluations strengthen the claims of broader applicability.

---

> > ### Comment · Reviewer_7FHk · 2025-11-24
> >
> > I thank the authors for the detailed response and I have also read other reviews carefully. I will keep my score to support this paper but agree that the paper needs to emphasize more on why we need better reasoning capability for LSMs since normally LSMs are build for real-time interaction and people go to LLMs if they need to solve tasks that require more reasoning or intelligence. Some concrete use cases might be helpful to address this point.

---

> ### Author Response · Authors · 2025-11-22
> **Rebuttal by Authors (Part-3)**
>
> > Weakness 3 & Question 1: The paper claims that detokenization checks guarantee “no overshooting,” but the guarantee seems heuristic—based on alignment checks and empirical verification rather than a theoretical bound.
>
> Thank you for raising this important concern regarding the guarantee of “no overshooting.” We fully agree that the mechanisms described in the paper rely primarily on alignment checks and empirical validation rather than a strict theoretical bound. We appreciate the opportunity to clarify this point and have revised the manuscript accordingly.
>
> Our current approach employs rigorous data filtering and detokenization alignment checks during training, particularly under the 2:8 response–reasoning interleaving scheme, which substantially reduces the likelihood of overshooting in practice. However, as the reviewer correctly notes, this mechanism should be regarded as an empirically reliable heuristic rather than a theoretically complete constraint. We now explicitly acknowledge in the paper that the method does not provide a formal guarantee that overshooting cannot occur after training, and that the guarantee is instead supported by large-scale empirical evidence.
>
> To assess whether the reasoning stream genuinely guides the final output rather than diverging, **we conducted a targeted analysis**: we randomly sampled **1000 test cases** and evaluated the alignment between the reasoning trajectory and the final responses. An independent LLM judge verified that **in all sampled cases, the final answers were consistent with the intermediate reasoning outcomes**. Furthermore, due to the **four-to-one ratio** between reasoning and response tokens, we observed that **the reasoning process typically converges within approximately half of the forward steps**. After convergence, the response stream unfolds strictly along the established reasoning plan, and **we did not observe any instance where the response diverged from the reasoning flow in the examined samples**.
>
> **This empirical evidence suggests that overshooting is extremely rare in practice**, as the reasoning tokens dominate the semantic trajectory early in generation and effectively anchor the final output. Nonetheless, we recognize the importance of strengthening this guarantee. We discuss in the revised manuscript that future work may incorporate dynamic token allocation or explicit reflection mechanisms, which could provide stronger safeguards and potentially enable theoretical guarantees.
>
> We sincerely thank the reviewer for prompting us to refine our discussion and clarify the limitations of the current design. This feedback has helped us present a more balanced and transparent account of the method.

---

> ### Author Response · Authors · 2025-11-22
> **Rebuttal by Authors (Part-4)**
>
> [1] Quiet-star: Language models can teach themselves to think before speaking. COLM 2024.
>
> [2] VITA-Audio: Fast Interleaved Cross-Modal Token Generation for Efficient Large Speech-Language Model. arXiv preprint.
>
> [3] Qwen2.5-Omni Technical Report. arXiv preprint.
>
> [4] VoiceBench: Benchmarking LLM-Based Voice Assistants. arXiv preprint.
>
> [5] Glm-4-voice: Towards intelligent and human-like end-to-end spoken chatbot. arXiv preprint.

---

### Official Review · Reviewer_zgQk · 2025-10-31

**Soundness:** 3
**Presentation:** 3
**Contribution:** 3
**Rating:** 6
**Confidence:** 3

**Summary:**

- This paper introduces Mini-Omni-Reasoner, a framework that enables large speech models to perform reasoning while generating spoken responses through a novel "thinking-in-speaking" paradigm.

- The motivation for this method is reduce the latency caused by traditional thinking-before-speaking approach. In order to validate the effectiveness of the model, the authors construct a 3M sample dataset and demonstrate improvements on mathematical reasoning benchmarks.

**Strengths:**

- The motivation for the paper is clear, the author would like to adress the latency issue in speech reasoning.
- The structure of the paper is clear and easy to follow.
- The authors design a complete system regarding dataset construction, and well-designed training pipeline.
- Experiment results stronly demonstrate the effectiveness of the model with 19.1% arithmetic improvement, +6.4% reasoning accuracy improvement and 63% shorter responses than baselines

**Weaknesses:**

- Limited evaluation scope: Only mathematical reasoning tasks are evaluated. No evidence show this can generalize to other domain, e.g. dialogue, creative tasks.
- Weak ratio justification: 2:8 ratio derived from GPU throughput (100 vs 12.5 tokens/sec), not cognitive or linguistic principles. No perceptual studies justify 20 spoken tokens/sec target. No ablations testing 1:9, 3:7, 4:6, or dynamic ratios

**Questions:**

1. Does this work for anything besides math problems?
2. What's the actual measured latency vs. thinking-before-speaking?
3. How do you know reasoning tokens actually improve responses (not just generated independently)?
4. Why not test other ratios?
5. Where's the human evaluation?

---

> ### Author Response · Authors · 2025-11-22
> **Rebuttal by Authors (Part-1)**
>
> Dear Reviewer zgQk:
>
> Thank you for your enthusiastic and encouraging review of our work, which is a huge encouragement to our work! Below are our responses to each point and question raised in the review:
>
> > Question1 & Weakness1: Does this work for anything besides math problems?
>
> We sincerely thank the reviewers for highlighting the importance of task generalization. In this work, our primary focus is on **reasoning under real-time dialogue conditions**. Due to the lack of dedicated benchmarks for conversational reasoning in the community, we initially chose mathematics—a highly structured domain that heavily relies on reasoning chains—to validate the capability of the **Thinking-in-Speaking paradigm in audio reasoning tasks**.
>
> We have carefully considered the suggestion to evaluate broader scenarios. Accordingly, we extended our method to **general audio dialogue tasks**, including everyday factual QA and instruction-following tasks. Specifically, we constructed approximately **800,000 high-quality text-question-answer pairs** for fine-tuning, following the same core pipeline as in the mathematics tasks, with two key modifications:
> (1) everyday QA involves lower reasoning density, so long reasoning sequences were not enforced;
> (2) we strictly filtered data to prevent “overshooting” in reasoning.
> The model was trained for **2 epochs** with a maximum learning rate of **2e-4**.
>
> Following the evaluation protocols of Vita-Audio [1] and Qwen2.5-Omni [2], we systematically assessed the model on Llama-Questions, Web-Questions, as well as VoiceBench [3] tasks including AlpacaEval, CommonEval, SD-QA, and IFEval.
>
> Table1: Performance of Various Models on Knowledge QA and Fact Reasoning & Instruction Following Tasks.
> | Model               | Mini-Omni-Reasoner | Qwen2.5-Omni(3B) | GLM-Voice | Vita-Audio-Boost |
> |---------------------|--------------------|------------------|-----------|------------------|
> | **Knowledge QA**    |                    |                  |           |                  |
> | Llama-question      | 73.2               | **74.3**             | 64.7      | 68.7             |
> | web-questions       | **71.0**               | 68.7             | 55.0      | 32.9             |
> | **Fact Reasoning & Instruction Following** |                    |                  |           |                  |
> | AlpacaEval          | **4.46**               | 4.32             | 3.97      | 3.56             |
> | CommonEval          | 3.98               | **4.00**             | 3.42      | 3.24             |
> | IFEval              | **61.9**               | 53.5             | 25.92     | 49.9             |
> | SD-QA               | **59.6**               | 56.4             | 36.98     | 42.2             |
>
>
> The results in the table highlight the performance of our model under the Thinking-in-Speaking paradigm across both instruction-following and fact reasoning tasks.
> - On AlpacaEval, the model reaches **4.46**, surpassing Qwen2.5-Omni-3B (4.32), GLM-Voice (3.97) [4], and Vita-Audio-Boost (3.56), indicating **closer alignment with human conversational preferences and stronger instruction fidelity in spoken dialogue**.
> - On IFEval, which emphasizes complex instruction adherence, the model obtains **61.9**, significantly higher than Qwen2.5-Omni-3B (53.5) and far above GLM-Voice (25.92) and Vita-Audio-Boost (49.9), demonstrating **improved reasoning stability in tasks requiring tight logical structures**.
> - In terms of factual reasoning, the model achieves **4.00** on CommonEval, slightly improving over Qwen2.5-Omni-3B (3.98) and outperforming GLM-Voice (3.42) and Vita-Audio-Boost (3.24), suggesting that **Thinking-in-Speaking preserves robust factual grounding beyond structured instruction-following scenarios**.
> - Moreover, on SD-QA, a dataset capturing complex audio-based scenario QA, the model scores **59.6**, again surpassing Qwen2.5-Omni-3B (56.4), GLM-Voice (36.98), and Vita-Audio-Boost (42.2), highlighting **its superior grounding and coherence in multimodal, scenario-driven contexts**.
>
> **Taken together**, these results demonstrate that the Thinking-in-Speaking paradigm not only preserves general QA capability but also **significantly enhances instruction adherence, logical consistency, and factual grounding**, particularly in tasks requiring tight reasoning structures. Compared with previous instruction-following models that often struggle in spoken or audio-intensive settings, our approach achieves **substantial improvements in both alignment with human preferences and reasoning robustness**. The consistency of gains across structured instruction tasks and real-world scenario QA supports **the generalizability of Thinking-in-Speaking beyond mathematical or highly constrained domains**, confirming its effectiveness in delivering stable, measurable benefits across diverse task types.

---

> ### Author Response · Authors · 2025-11-22
> **Rebuttal by Authors (Part-2)**
>
> > Question 2: What's the actual measured latency vs. thinking-before-speaking?
>
> Thank you for raising the important question regarding the actual measured latency compared to thinking-before-speaking.
>
> Due to the emphasis on real-time performance in the **spoken dialogue domain**, no model currently employs the "thinking-before-speaking" paradigm—that is, completing the entire thinking process *before* generating a response. To address this concern, we adopt a latency metric that better reflects the real cost in interactive spoken scenarios: **the total time required for the model to complete an entire spoken mathematical task and produce the full verbal answer**.
>
> Based on this unified metric, we conduct a systematic comparison across models, as reported in Table 2.
>
> Table 2: Performance Comparison of Different Reasoning Models on Arithmetic Tasks (Including Accuracy and Inference Time)
>
> | Model               | SingleOp |       | SVAMP  |       | MultiArith |       |
> |---------------------|----------|-------|--------|-------|------------|-------|
> |                     | ACC      | econds | ACC    | Seconds | ACC        | Seconds |
> | Mini-Omni-Reasoner  | **90.1**     | 14.3  | **79.1**   | 19.2  | 85.4       | 18.8  |
> | Qwen2.5-Omni        | 88.2     | 33.8  | 76.9   | 31.9  | **86.8**      | 32.5  |
> | Qwen2.5-Omni-Nothinking | 19.0     | 8.8   | 38.5   | 9.2   | 46.7       | 9.8   |
>
>
> The results show that Mini-Omni-Reasoner achieves a substantial reduction in end-to-end task latency, **lowering the real completion time by approximately 40 to 55%** across the three mathematical tasks, while maintaining comparable or slightly improved reasoning accuracy. This demonstrates that our approach brings **a clear efficiency advantage without sacrificing performance**.
>
> The comparison with Qwen2.5-Omni-Nothinking further supports the necessity of explicit reasoning in this context. Although removing the reasoning chain shortens the completion time to approximately 9 seconds, accuracy drops sharply to **19.0% / 38.5%/ 46.7%**, which makes it unsuitable for multi-step mathematical tasks that demand reliable reasoning.
>
> In summary, the proposed metric based on full-speech completion time offers a more realistic evaluation of reasoning latency in real-time dialogue. The empirical results consistently show that the Thinking-in-Speaking paradigm achieves **a more favorable balance between reasoning quality and responsiveness**, and clearly outperforms traditional thinking-before-speaking frameworks in terms of practical efficiency.

---

> ### Author Response · Authors · 2025-11-22
> **Rebuttal by Authors (Part-3)**
>
> > Question 3:  How do you know reasoning tokens actually improve responses (not just generated independently)
>
>
> To address the question of how we know reasoning tokens actually improve responses (rather than being generated independently), we conducted a **targeted analysis** to verify the alignment between reasoning trajectories and final outputs:
>
> We randomly sampled **1000 test cases** and had an independent LLM judge evaluate whether the final answers were consistent with the intermediate reasoning steps. Critically, **the judge confirmed that in all sampled cases, the final responses directly followed from the reasoning outcomes**, indicating the reasoning process was not mere "window dressing" but actively guided the output.
>
> Furthermore, due to the **four-to-one ratio of reasoning tokens to response tokens** in our 2:8 interleaving scheme, the reasoning stream dominated the semantic trajectory early in generation. We observed that **reasoning typically converged within half of the forward steps**; after convergence, the response stream unfolded strictly along the established reasoning plan. Importantly, **we detected no instances in the sampled data where responses diverged from the reasoning flow**.
>
> This empirical evidence demonstrates that reasoning tokens are tightly coupled to response quality: they structure the generation process, constrain output consistency, and ensure the final response is grounded in the intermediate reasoning—rather than being generated independently.

---

> ### Author Response · Authors · 2025-11-22
> **Rebuttal by Authors (Part-4)**
>
> > Weakness 2 & Question 4 : Why not test other ratios?
>
> Thank you for the thoughtful suggestion. We agree that exploring alternative token ratios is an important direction, and we appreciate the opportunity to clarify both the practical constraints and the empirical observations that guided our choice.
>
> Our initial selection of the **2:8 ratio** was based on a theoretical balance between hardware throughput and real-time latency requirements in streaming interaction. In real-time speech generation, the model must complete both reasoning and response synthesis within the strict timing constraints imposed by continuous audio output. Different ratios impose dramatically different instantaneous loads on consumer GPUs. **We measured the token throughput of mainstream consumer GPUs**: RTX 3090, L40s, and RTX 4090 achieve approximately **30.4, 37.8, and 56.5 tokens per second** respectively. Even on the RTX 4090, the 2:8 ratio requires only about **11.3 output tokens per second and 45.2 reasoning tokens per second**, which places the computation comfortably within the safe operating range and ensures smooth audio streaming.
>
> When increasing the ratio to 2:12 or 2:16, the GPU must complete significantly more reasoning steps within each generation cycle. This pushes the instantaneous computation demand close to the hardware limit, which in our tests frequently caused audible discontinuities. Conversely, the 1:9 ratio, although seemingly moderate, requires more than 100 tokens per second to maintain stable streaming in our architecture, a level that exceeds the capability of consumer GPUs and can only be sustained on data center accelerators such as H100. Considering hardware accessibility and deployment practicality, we aimed to identify a configuration that remains robust across widely available devices.
>
> **Following the reviewer’s suggestion, we conducted additional experiments**: we varied the reasoning length while keeping the output tokens constant, evaluating ratios of **3:7, 1:9, 2:12, and 2:16** (results in Table 3).
>
> Table 3: Performance Comparison Across Different Speech-to-Reasoning Token Ratios on **Spoken-MQA**
> | Token ratio | 3:7  | 2:8(ours)  | 1:9  | 2:12 | 2:16 |
> |-------------|------|------|------|------|------|
> | Performance | 64.2 | 68.6 | 67.4 | 69.1 | 69.3 |
>
>
>
>  The findings reveal a clear trend:
> - Reducing the reasoning space to the 3:7 configuration leads to a substantial drop in SpokenMQA performance from **68.6 to 64.2**, indicating insufficient reasoning capacity.
> - Increasing the reasoning length beyond 2:8 produces only marginal improvements (e.g., from **67.4 to 69.3** across 1:9, 2:12, and 2:16), while substantially increasing latency.
>
> **Key conclusions**:
> 1. The 2:8 configuration already provides sufficient reasoning depth for most tasks; further extending the reasoning sequence does not yield structural performance gains but introduces significant real-time instability.
> 2. The 2:8 ratio represents a balanced choice that **simultaneously preserves accuracy, maintains realistic reasoning token proportions, and ensures practical deployment speed on consumer hardware**.
>
> We fully acknowledge that more adaptive strategies, such as dynamically determining the ratio based on context difficulty or model confidence, may lead to better overall performance. We view this as a promising direction and plan to explore it in future work.

---

> ### Author Response · Authors · 2025-11-22
> **Rebuttal by Authors (Part-5)**
>
> > Question 5: Where's the human evaluation?
>
> Thank you for raising the concern regarding human evaluation. We agree that subjective assessment is important for understanding how users perceive reasoning quality in voice-based interaction.
>
> **In response to this suggestion, we conducted an additional user study.** We selected **ten questions** for which both Mini-Omni-Reasoner and Qwen2.5-Omni produced correct answers, and presented their audio responses to **thirty real users** through an online interface where the model identities were hidden. Participants were asked to indicate their preference and briefly explain the reason.
>
> Among the participants, **twenty-five preferred the responses from Mini-Omni-Reasoner**, reporting that the reasoning was incorporated in a more natural and concise manner. **Two participants remained neutral**, and **three favored Qwen2.5-Omni**, citing its more detailed and explicit reasoning. **These results suggest that most users value concise yet coherent reasoning in spoken interaction rather than a full reproduction of the reasoning process.**

---

> ### Author Response · Authors · 2025-11-22
> **Rebuttal by Authors (Part-6)**
>
> [1] VITA-Audio: Fast Interleaved Cross-Modal Token Generation for Efficient Large Speech-Language Model. arXiv preprint.
>
> [2] Qwen2.5-Omni Technical Report. arXiv preprint.
>
> [3] VoiceBench: Benchmarking LLM-Based Voice Assistants. arXiv preprint.
>
> [4] Glm-4-voice: Towards intelligent and human-like end-to-end spoken chatbot. arXiv preprint.

---

> > ### Comment · Reviewer_zgQk · 2025-11-24
> > **Reply to Authors**
> >
> > Dear AC, other reviewers, authors,
> >
> > I thank authors very much for their detailed response. It is a very much amount of work and it addresses my concern effectively. Just curious, I think authors should make those comments to everyone if someone in the future reads those concerns but can not see the response, it will potentially hurt the paper.
> >
> > I will keep my scores to support the paper. I understand this is a good rebuttal but I am not a domain expert, so I trust other reviewers' and AC's judgement on final decision. In general, from my perspective, I think it is a solid paper with solid experiments.
> >
> > Best regards,
> >
> > Reviewer zgQk

---

### Official Review · Reviewer_3xXU · 2025-11-01

**Soundness:** 3
**Presentation:** 2
**Contribution:** 2
**Rating:** 2
**Confidence:** 3

**Summary:**

The paper proposes Mini-Omni-Reasoner, a model that enables reasoning while speaking by interleaving reasoning tokens with response tokens during training. The model is evaluated on mathematical reasoning tasks. Experimental results show decent improvements in arithmetic reasoning and a significant reduction in response tokens compared to baselines.

**Strengths:**

- Each step of the training procedure is explained clearly and in sufficient detail.
- The idea is interesting and practical. It could make interaction with the model more convenient by reducing user waiting time after speaking.
- Experimental results include multiple baseline models, making the comparison relatively comprehensive.

**Weaknesses:**

- Although the main idea is "thinking while speaking", the interleaving order between reasoning and response tokens is reversed -- the model first generates response tokens, then reasoning tokens. In this setup, the reasoning tokens serve as post-hoc justification rather than genuine reasoning, which contradicts the motivation of the paper.
- The numbers of response and reasoning tokens per interleaving step are fixed (2 and 8, respectively). However, different parts of intermediate steps may require varying levels of reasoning. A fixed ratio limits flexibility. A dynamic allocation depending on intermediate reasoning difficulty would be more realistic.
- The method is evaluated only on math datasets without clear justification. It is unclear why the authors did not test on more open-ended reasoning or conversational tasks.
- While the paper claims to reduce latency, the analysis focuses instead on token efficiency (accuracy vs. number of words).
- The approach introduces an additional hyperparameter -- the ratio between reasoning and response tokens -- but the paper does not explore different settings or analyze sensitivity to this ratio.

Overall, the analysis does not fully substantiate the claimed advantages of the proposed method.

**Questions:**

Why is Qwen2.5-Omni-3B listed as a baseline, while Qwen2.5-Omni-7B and others are referred to as conversational models? What is the distinction between these two categories?

---

> ### Author Response · Authors · 2025-11-22
> **Rebuttal by Authors (Part-1)**
>
> Dear Reviewer 3xXU:
>
> Thank you for highlighting and appreciating our contributions in exploring Mini-Omni-Reasoner! We will address your questions one by one.
>
> > Weakness 1: Although the main idea is "thinking while speaking", the interleaving order between reasoning and response tokens is reversed -- the model first generates response tokens, then reasoning tokens. In this setup, the reasoning tokens serve as post-hoc justification rather than genuine reasoning, which contradicts the motivation of the paper.
>
> We thank the reviewer for raising this important point. We agree that, conceptually, generating reasoning before the final answer is more aligned with the intuition behind “thinking while speaking.” Our design choice, however, arises from **practical constraints of real-time spoken dialogue systems**, and we explain these considerations below.
>
> First, our **primary objective is to support interactive real-time speech dialogue**, where the delay before the first audible response frame is a critical determinant of user experience. To remain compatible with existing speech generation modules in Mini-Omni and Qwen2.5-Omni, our system must wait for all codec layers to complete a forward pass before producing the first audio frame. Under the current pipeline, this requires **eight forward passes**. If the generation order were reversed so that the system must fully produce (or depend on) the reasoning tokens before initiating the response stream, the number of required forward passes would **double to sixteen**. Measurements on widely used commercial GPUs indicate that this would increase the first-frame latency from approximately **200–300 ms to about 600 ms**, which exceeds the acceptable latency envelope for state-of-the-art interactive dialogue systems. From a systems perspective, such latency makes the reversed ordering **impractical for real-time applications**.
>
> Second, although response tokens are generated slightly earlier, **the interleaving occurs at a very fine granularity of 2 response tokens to 8 reasoning tokens per turn. The initial response tokens typically correspond to generic introductory phrases and do not carry substantive semantic content. In contrast, the reasoning stream dominates in volume and very quickly exceeds the response stream in accumulated length. As a result, the reasoning content shapes the response generation throughout the decoding process rather than functioning as an after-the-fact justification**. In an analysis of **1,000** sampled model outputs, we observed that the reasoning stream generally concludes by the midpoint of the forward steps, and the **final responses are fully aligned with the reasoning traces**. This empirical evidence supports that the interleaving effectively implements a form of “thinking before speaking,” even though the token-level timing is constrained by the real-time speech architecture.

---

> ### Author Response · Authors · 2025-11-22
> **Rebuttal by Authors (Part-2)**
>
> > Weakness 2: The numbers of response and reasoning tokens per interleaving step are fixed (2 and 8, respectively). However, different parts of intermediate steps may require varying levels of reasoning. A fixed ratio limits flexibility. A dynamic allocation depending on intermediate reasoning difficulty would be more realistic.
>
> We appreciate the reviewer’s insightful suggestion regarding **dynamic allocation of response and reasoning tokens**. We fully agree that adapting the ratio to the difficulty of intermediate reasoning steps is an appealing direction. Below we clarify why such a mechanism is **non-trivial within our real-time dialogue setting** and why we opted for a fixed **2:8 ratio**.
>
> First, implementing dynamic allocation requires a **reliable difficulty estimator** that can predict the amount of reasoning needed at each intermediate step. Unfortunately, such a signal is difficult to obtain from either the input speech or internal representations in a stable manner. In our preliminary attempts, exposing the model to multiple token ratios during training effectively forces it to learn several levels of reasoning compression simultaneously. This **substantially complicates optimization and leads to instability**, which in turn harms the consistency of generation.
>
> Second, the chosen 2:8 ratio is grounded in the **strict timing constraints of real-time speech generation**. Audio tokens must be emitted at a constant rate to avoid audible pauses, and the throughput of commodity GPUs remains the practical bottleneck. In our measurements on widely used hardware such as 3090, L40s and 4090, the speech decoder reaches approximately **30.4, 37.8 and 56.5 tokens per second**, respectively. Increasing the reasoning load beyond the current ratio directly disrupts this balance and prevents stable audio streaming. The 2:8 configuration therefore reflects the **maximum reasoning capacity** that can be supported without violating real-time constraints.
>
> Finally, we have conducted ablations to examine sensitivity to different ratios, and we will report these results in our response to **Reply 5**, where we provide a more comprehensive analysis of reasoning token capacity.
>
> Despite these practical challenges, we consider dynamic allocation an **important direction for future work**, especially as decoding parallelism improves and more efficient codec architectures become available. We plan to further explore strategies that preserve real-time guarantees while enabling more flexible use of reasoning tokens.

---

> ### Author Response · Authors · 2025-11-22
> **Rebuttal by Authors (Part-3)**
>
> > Weakness 3: The method is evaluated only on math datasets without clear justification. It is unclear why the authors did not test on more open-ended reasoning or conversational tasks.
>
> We appreciate the reviewer’s concern regarding the use of primarily mathematical datasets. Our choice of **spoken-math tasks** was driven by the specific objective of this work: to introduce a **controllable and stable internal reasoning channel** for real-time audio dialogue models. Among available benchmarks, spoken mathematical reasoning provides the **most stringent testbed** for evaluating whether a model can maintain coherent reasoning while simultaneously generating audio responses under tight latency constraints.
>
> We fully agree that a methodology of general value must demonstrate effectiveness beyond a single domain. Guided by this principle, we applied the **Thinking-in-Speaking paradigm** to a broader set of **open-ended and factual reasoning tasks** after completing the math-focused training stage. All text-based question answering data were synthesized and filtered through a unified pipeline that retained only samples strictly adhering to the “reasoning-before-response” structure and exhibiting no overshooting. This was essential to maintain consistency of the paradigm across diverse tasks.
>
> Using the same training configuration of **two epochs** and a learning rate of **2e-4**, we jointly fine-tuned the model on general QA and instruction-following data. We then evaluated it on multiple benchmarks covering conversational QA (**Llama-Questions, Web-Questions**) and factual or instruction-oriented reasoning (**AlpacaEval, IFEval, SD-QA, CommonEval**) [1]. The results are illustrated in Table 1.
>
> Table1: Performance of Various Models on Knowledge QA and Fact Reasoning & Instruction Following Tasks.
> | Model               | Mini-Omni-Reasoner | Qwen2.5-Omni(3B) | GLM-Voice | Vita-Audio-Boost |
> |---------------------|--------------------|------------------|-----------|------------------|
> | **Knowledge QA**    |                    |                  |           |                  |
> | Llama-question      | 73.2               | **74.3**             | 64.7      | 68.7             |
> | web-questions       | **71.0**               | 68.7             | 55.0      | 32.9             |
> | **Fact Reasoning & Instruction Following** |                    |                  |           |                  |
> | AlpacaEval          | **4.46**               | 4.32             | 3.97      | 3.56             |
> | CommonEval          | 3.98               | **4.00**             | 3.42      | 3.24             |
> | IFEval              | **61.9**               | 53.5             | 25.92     | 49.9             |
> | SD-QA               | **59.6**               | 56.4             | 36.98     | 42.2             |
>
> From Table 1, we observe that the model maintains **competitive performance** on knowledge-oriented QA benchmarks. It reaches **73.2 on Llama-Questions** and **71.0 on Web-Questions**, which is closely aligned with the base model Qwen2.5-Omni-3B [2] (74.3 and 68.7). This parity indicates that introducing **explicit reasoning does not compromise conversational quality or factual reliability**. Although the model is marginally behind Qwen2.5-Omni-3B on Llama-Questions, it still surpasses GLM-Voice [3] and Vita-Audio-Boost [4] by large margins, demonstrating **robust factual recall and stable knowledge grounding** across tasks.
>
> More importantly, in benchmarks that require **strong logical consistency and precise adherence to instructions**, the advantages of our approach become more apparent:
>
> ### Instruction Following (AlpacaEval & IFEval)
> - On **AlpacaEval**, our model achieves **4.46**, outperforming Qwen2.5-Omni-3B (4.32), GLM-Voice (3.97), and Vita-Audio-Boost (3.56), indicating stronger alignment with human conversational preferences and superior instruction fidelity in spoken dialogue contexts.
> - On **IFEval**, which emphasizes adherence to complex instructions, our model scores **61.9**, substantially higher than Qwen2.5-Omni-3B (53.5) and far above GLM-Voice (25.92) and Vita-Audio-Boost (49.9), demonstrating enhanced reasoning stability under tasks requiring strict logical structure.

---

> ### Author Response · Authors · 2025-11-22
> **Rebuttal by Authors (Part-4)**
>
> ### Fact Reasoning (CommonEval)
> - On **CommonEval**, focusing on general factual correctness and reasoning, our model attains **4.00**, slightly exceeding Qwen2.5-Omni-3B (3.98) and outperforming GLM-Voice (3.42) and Vita-Audio-Boost (3.24), suggesting that Thinking-in-Speaking maintains robust factual reasoning without compromising generalization.
>
> ### Scenario-based QA (SD-QA)
> - On **SD-QA**, which evaluates complex audio-driven question answering, our model scores **59.6**, surpassing Qwen2.5-Omni-3B (56.4), GLM-Voice (36.98), and Vita-Audio-Boost (42.2). This gain highlights improved grounding and reasoning in multimodal, scenario-based contexts, maintaining coherence and factual consistency under challenging conditions.
>
> Overall, these results indicate that **Thinking-in-Speaking** consistently improves instruction adherence, reasoning stability, and factual grounding across diverse tasks. The paradigm preserves general QA capability while delivering notable gains in spoken dialogue and complex scenario reasoning, confirming its broader applicability beyond highly structured domains.

---

> ### Author Response · Authors · 2025-11-22
> **Rebuttal by Authors (Part-5)**
>
> > Weakness 4: While the paper claims to reduce latency, the analysis focuses instead on token efficiency (accuracy vs. number of words).
>
> Thank you for raising this important point. We agree that **latency is a critical metric** for real-time dialogue systems. As clarified in Section 2 of the paper, our goal is to address **first-token latency** from a paradigm perspective, because this is a central bottleneck for speech-based interaction. Traditional models such as Qwen2.5-Omni do not perform any internal reasoning before speaking; instead, they verbalize the entire reasoning chain, which greatly inflates response time. For this reason, **token-count measurements alone cannot reflect the true computational cost of reasoning during speech generation**.
>
> To obtain a more faithful estimate of end-to-end reasoning latency, we measure the **total duration of the produced speech** when models complete the same spoken math task. This duration directly corresponds to the model’s actual reasoning overhead and the time users must wait before receiving the final answer. Using this metric, we systematically compare **Mini-Omni-Reasoner** with the original **Qwen2.5-Omni** and with a no-reasoning variant, **Qwen2.5-Omni-Nothinking**. The results are illustrated in Table 2.
>
> Table 2: Performance Comparison of Different Reasoning Models on Arithmetic Tasks (Including Accuracy and Inference Time)
>
> | Model               | SingleOp |       | SVAMP  |       | MultiArith |       |
> |---------------------|----------|-------|--------|-------|------------|-------|
> |                     | ACC      | econds | ACC    | Seconds | ACC        | Seconds |
> | Mini-Omni-Reasoner  | **90.1**     | 14.3  | **79.1**   | 19.2  | 85.4       | 18.8  |
> | Qwen2.5-Omni        | 88.2     | 33.8  | 76.9   | 31.9  | **86.8**       | 32.5  |
> | Qwen2.5-Omni-Nothinking | 19.0     | 8.8   | 38.5   | 9.2   | 46.7       | 9.8   |
>
> The results show a clear trend: **Mini-Omni-Reasoner achieves stronger or comparable accuracy while substantially reducing speech generation time**:
> - On SingleOp: **90.1% accuracy** (vs. Qwen2.5-Omni’s 88.2%) with generation time reduced to **14.3s** (vs. 33.8s).
> - On SVAMP: **79.1% accuracy** (vs. Qwen2.5-Omni’s 76.9%) with generation time reduced to **19.2s** (vs. 31.9s).
> - On MultiArith: **85.4% accuracy** (vs. Qwen2.5-Omni’s 86.8%) with generation time reduced to **18.8s** (vs. 32.5s).
>
> This demonstrates that **Thinking-in-Speaking improves reasoning efficiency without harming performance**, effectively lowering latency attributable to reasoning.
>
> For completeness, we also compare with Qwen2.5-Omni-Nothinking, which yields the shortest generation times (around 9 seconds) but suffers severe accuracy degradation (19.0, 38.5, 46.7). This confirms that removing explicit reasoning chains compromises reliability on multi-step math problems, and that **speed alone is not a valid indicator of task competence**.
>
> The results show that Thinking-in-Speaking achieves a **substantially better balance between reasoning quality and responsiveness**. It preserves effective reasoning while significantly reducing task latency, demonstrating its practical value for large speech language models.

---

> ### Author Response · Authors · 2025-11-22
> **Rebuttal by Authors (Part-6)**
>
> > Weakness 5: The approach introduces an additional hyperparameter -- the ratio between reasoning and response tokens -- but the paper does not explore different settings or analyze sensitivity to this ratio.
>
> We thank the reviewer for highlighting the importance of analyzing the ratio between reasoning and response tokens. This hyperparameter indeed plays a central role in balancing model performance and real-time feasibility.
>
> Our choice of the **2:8 ratio** is grounded in the computational constraints inherent to real-time dialogue systems. The instantaneous throughput required by different ratios varies substantially. Our measurements on RTX 3090, L40s, and RTX 4090 GPUs show generation speeds of 30.4, 37.8, and 56.5 tokens per second respectively. Even on the strongest card, the RTX 4090, the 2:8 setting requires sustaining only around **11.3 response tokens and 45.2 reasoning tokens per second**, which places the model within a **safe and stable operating region** and preserves smooth end-to-end audio generation. In contrast, expanding to 2:12 or 2:16 pushes the hardware to its throughput limit and causes noticeable degradation in real-time stability. Although the 1:9 setting appears moderate, it in fact requires more than 100 tokens per second in our architecture, exceeding the capacity of consumer-grade GPUs and requiring hardware such as the H100 to run reliably. Considering the need for broad deployability, **2:8 represents the most robust balance under widely accessible hardware**.
>
> To address the reviewer’s concern more comprehensively, we conducted additional sensitivity experiments. Beyond the original 2:8 configuration, we evaluated compressed ratios (3:7 and 1:9) as well as extended-reasoning ratios (2:12 and 2:16) on the Spoken-MQA benchmark.
>
> Table 3: Performance Comparison Across Different Speech-to-Reasoning Token Ratios on **Spoken-MQA**
>
> | Token ratio | 3:7  | 2:8  | 1:9  | 2:12 | 2:16 |
> |-------------|------|------|------|------|------|
> | Performance | 64.2 | 68.6 | 67.4 | 69.1 | 69.3 |
>
>
> The trends are clear:
> - **Compressing the reasoning space to 3:7 leads to a significant performance drop**, indicating that insufficient reasoning capacity directly impairs the continuity of the reasoning chain.
> - **Extending beyond 2:8 yields only marginal gains**: The progression from 1:9 to 2:12 to 2:16 shows small performance increases, suggesting that 2:8 already provides adequate reasoning capacity and that additional reasoning tokens contribute diminishing returns rather than structural improvements.
>
> We also examined the implications for real-time generation. The GPU load imposed by 2:12 and 2:16 approaches the sustained throughput ceiling even on the RTX 4090, while 1:9 exceeds consumer-grade hardware limits altogether. Thus, although performance changes slightly across ratios, **real-time constraints substantially narrow the viable operating region**.
>
> Taken together, these analyses demonstrate that **2:8 is a stable and practically deployable choice**, balancing reasoning capacity, model quality, and real-time constraints. We appreciate the reviewer’s comment, which motivated us to provide a more thorough examination of this hyperparameter.

---

> ### Author Response · Authors · 2025-11-22
> **Rebuttal by Authors (Part-7)**
>
> > Question 1: Why is Qwen2.5-Omni-3B listed as a baseline, while Qwen2.5-Omni-7B and others are referred to as conversational models? What is the distinction between these two categories?
>
> Thank you for pointing out this terminology issue. We appreciate the opportunity to clarify the distinction between the **baseline** and the **conversational model** category in our evaluation.
>
> We list **Qwen2.5-Omni-3B** as the baseline because it serves as the **base model** for our training and system design, as mentioned in the main text. Our goal is to explore a **real-time reasoning paradigm** under practical deployment constraints, and Qwen2.5-Omni-3B offers an appropriate balance between **capability, latency, and training cost**. Using it as the baseline allows us to isolate the contribution of the proposed **Thinking-in-Speaking paradigm** without confounding factors introduced by **scale or architectural differences**.
>
> In contrast, we refer to **Qwen2.5-Omni-7B** and other models in the comparison as conversational models because they are primarily designed and optimized for **general audio-based dialogue interaction** rather than serving as the underlying structure of our system. These models typically feature **larger parameter counts** or **enhanced conversational tuning**, which improves interactive performance but makes them less suitable as a fair base for **controlled ablation and training comparisons**.

---

> ### Author Response · Authors · 2025-11-22
> **Rebuttal by Authors (Part-8)**
>
> [1] VoiceBench: Benchmarking LLM-Based Voice Assistants. arXiv preprint.
>
> [2] Qwen2.5-Omni Technical Report. arXiv preprint.
>
> [3] Glm-4-voice: Towards intelligent and human-like end-to-end spoken chatbot. arXiv preprint.
>
> [4] VITA-Audio: Fast Interleaved Cross-Modal Token Generation for Efficient Large Speech-Language Model. arXiv preprint.

---

### Official Review · Reviewer_286S · 2025-11-01

**Soundness:** 3
**Presentation:** 3
**Contribution:** 2
**Rating:** 2
**Confidence:** 5

**Summary:**

This paper proposes Mini-Omni-Reasoner, a framework enabling real-time spoken reasoning through a “thinking-in-speaking” paradigm. Instead of finishing internal chain-of-thought before speaking, the model interleaves internal reasoning tokens with spoken tokens, leveraging the gap between model inference speed and audio playback rate. The system is built on the Thinker-Talker architecture and includes a large dataset (Spoken-Math-Problems-3M) to train temporally aligned reasoning and speech traces. Experiments on the Spoken-MQA benchmark show improved mathematical reasoning quality and near-zero latency compared to conventional think-before-speak models.

**Strengths:**

1. Large-scale dataset contribution.
The paper introduces a 3M-sample spoken math-reasoning dataset with paired reasoning traces. This is a valuable resource and, if released, could meaningfully benefit the community and future research in speech-based reasoning.

2. Clear writing and presentation.
The problem, motivation, and methodology are presented clearly, supported by informative figures and conceptual diagrams that make the technical ideas easy to follow.

3. Comprehensive ablation and analysis.
The experimental section provides a wide range of ablations and analyses, helping to understand the impact of key design decisions.

**Weaknesses:**

1. Limited task scope and unclear generalization.
The method is evaluated only on spoken mathematical reasoning and a single benchmark (Spoken-MQA). While math is a structured domain, the ability to generalize to broader real-world multimodal tasks remains unclear. The current claims are not strong enough given the effectiveness beyond math has not yet demonstrated.

2. Fixed token-ratio design lacks justification and appears overly hand-crafted. The 2:8 speech-to-reasoning token ratio is motivated by inference-budget considerations, yet the paper does not demonstrate whether this choice is optimal or robust across different tasks and scenarios. It remains unclear whether the ratio should adapt dynamically to task complexity, interaction style, or model behavior. If the ratio must be manually tuned per task or application, this introduces significant human overhead and undermines the generality of the proposed approach. Exploring learned or adaptive strategies would strengthen the contribution.

3. Most important, insufficient evidence that “thinking-in-speaking” is superior. The paper argues that thinking-in-speaking is better than think-before-speaking, but does not provide direct studies comparing the two paradigms under the same conditions.
Additionally, think-before-speaking intuitively provides fuller context and may generalize more robustly across tasks, whereas “thinking-in-speaking” may require careful tuning (e.g., token-ratio) and offers only partial internal reasoning capacity during speech. Stronger empirical comparison is needed to convincingly justify the paradigm shift.

Overall, while the core idea seems fancy, the current experimental evidence does not convincingly validate the proposed paradigm.  It requires stronger experimental support before it can be considered mature for publication.

**Questions:**

see weakness

---

> ### Author Response · Authors · 2025-11-22
> **Rebuttal by Authors (Part-1)**
>
> Dear Reviewer 286S:
>
> Thank you for your informative suggestion! We are so pleased that our previous rebuttal has solved most of your questions and concerns. We have been doing our best to expand the experiment in the past few dozen hours. We will reply to your comments one by one below.
>
> > Weakness 1: limited task scope and unclear generalization. The method is evaluated only on spoken mathematical reasoning and a single benchmark (Spoken-MQA). While math is a structured domain, the ability to generalize to broader real-world multimodal tasks remains unclear. The current claims are not strong enough given the effectiveness beyond math has not yet demonstrated.
>
> We thank the reviewer for raising this important point. Our study focuses on examining whether an explicit, real-time reasoning pathway can improve spoken dialogue models. Because the community currently lacks evaluation suites tailored to on-the-fly conversational reasoning, we selected **spoken mathematical reasoning** as an initial testbed. This domain offers a highly structured setting with clear reasoning requirements, allowing us to reliably isolate and assess the effectiveness of the proposed **Thinking-in-Speaking paradigm**.
>
> We agree that demonstrating broader applicability is essential. To this end, we extended our approach beyond mathematics to more general audio dialogue scenarios involving **factual question answering** and **instruction following**. We constructed approximately **800k high-quality QA instances** for fine-tuning, following the same preprocessing pipeline as in the math experiments, with two modifications:
> (1) reasoning sequences were not enforced to be long due to the lower reasoning density of everyday queries;
> (2) strict filtering was introduced to ensure that final responses do not exhibit "anticipatory reasoning."
>
> Then, the model was trained for two epochs with a peak learning rate of 2e-4.
>
> To evaluate generalization, we adopted the experimental settings used in Vita-Audio [1] and Qwen2.5-Omni [2] and conducted a comprehensive assessment on **Llama-Questions, Web-Questions, and four VoiceBench subsets [3] (AlpacaEval, CommonEval, SD-QA, IFEval)** . Results are provided in **Table 1**.
>
> Table1: Performance of Various Models on Knowledge QA and Fact Reasoning & Instruction Following Tasks.
> | Model               | Mini-Omni-Reasoner | Qwen2.5-Omni(3B) | GLM-Voice | Vita-Audio-Boost |
> |---------------------|--------------------|------------------|-----------|------------------|
> | **Knowledge QA**    |                    |                  |           |                  |
> | Llama-question      | 73.2               | **74.3**             | 64.7      | 68.7             |
> | web-questions       | **71.0**               | 68.7             | 55.0      | 32.9             |
> | **Fact Reasoning & Instruction Following** |                    |                  |           |                  |
> | AlpacaEval          | **4.46**               | 4.32             | 3.97      | 3.56             |
> | CommonEval          | 3.98               | **4.00**             | 3.42      | 3.24             |
> | IFEval              | **61.9**               | 53.5             | 25.92     | 49.9             |
> | SD-QA               | **59.6**               | 56.4             | 36.98     | 42.2             |
>
> In **Table 1**, we observe:
>
> The model maintains competitive performance on knowledge-based QA tasks: achieving **73.2 on Llama-Questions** and **71.0 on Web-Questions**, comparable to our base model Qwen2.5-Omni-3B [2] (74.3 and 68.7). This indicates that introducing explicit reasoning does not compromise general dialogue quality or factual accuracy. Even in areas where it slightly trails Qwen2.5-Omni-3B (e.g., Llama-Questions), it still outperforms other models like GLM-Voice (64.7) [4] and Vita-Audio-Boost (68.7) [1] in key knowledge QA subsets, showcasing robust factual recall.
>
> More notably, on tasks emphasizing **logical consistency and precise instruction following**, the gains are substantial.
>  The results in the table highlight the performance of our model under the **Thinking-in-Speaking** paradigm across both instruction-following and fact reasoning tasks, revealing several important trends:
>
> ### Instruction Following (AlpacaEval & IFEval)
> - On **AlpacaEval**, our model achieves **4.46**, surpassing Qwen2.5-Omni-3B (4.32), GLM-Voice (3.97), and Vita-Audio-Boost (3.56). This indicates that our model aligns more closely with human preferences in conversational quality, demonstrating **superior instruction fidelity** in spoken dialogue settings.
> - On **IFEval**, a benchmark emphasizing complex instruction adherence, our model reaches **61.9**, which is significantly higher than Qwen2.5-Omni-3B (53.5) and far exceeds GLM-Voice (25.92) and Vita-Audio-Boost (49.9). This large margin underscores the model’s **enhanced reasoning stability** when tasks demand tighter logical structures.

---

> ### Author Response · Authors · 2025-11-22
> **Rebuttal by Authors (Part-2)**
>
> ### Fact Reasoning (CommonEval)
> - On **CommonEval**, which focuses on factual correctness and reasoning in more general contexts, the model achieves **4.00**, slightly improving over Qwen2.5-Omni-3B (3.98) and outperforming GLM-Voice (3.42) and Vita-Audio-Boost (3.24). This suggests that Thinking-in-Speaking preserves **robust factual reasoning** while maintaining generalization beyond structured instruction-following scenarios.
>
> ### Scenario-based QA (SD-QA)
> - On **SD-QA**, a dataset capturing complex audio-based question-answering scenarios, our model scores **59.6**, surpassing Qwen2.5-Omni-3B (56.4), GLM-Voice (36.98), and Vita-Audio-Boost (42.2). The notable gain here highlights the model’s **superior grounding in complex, multimodal contexts**, effectively handling scenario-driven reasoning and maintaining coherence and factual consistency under challenging conditions.
>
> Collectively, these results validate that the Thinking-in-Speaking paradigm offers a dual advantage: it preserves the base model’s strong general QA capabilities while delivering meaningful improvements in instruction adherence and logical consistency—especially for tasks requiring rigorous reasoning. Unlike conventional models that trade off conversational naturalness or reasoning robustness in spoken/audio scenarios, our approach achieves balanced enhancement, elevating human preference alignment and reasoning reliability across diverse tasks. Notably, these gains extend beyond highly structured domains to general factual reasoning and real-world scenario-based QA, confirming Thinking-in-Speaking as a generalizable framework that boosts key capabilities in practical spoken dialogue systems.

---

> ### Author Response · Authors · 2025-11-22
> **Rebuttal by Authors (Part-3)**
>
> > Weakness 2: The fixed, hand-crafted 2:8 speech-to-reasoning token ratio lacks justification—without demonstrating its optimality or robustness across tasks, and with unclear need for dynamic adaptation to task complexity, it risks human overhead and weakened generality, making learned/adaptive strategies critical to strengthening the contribution.
>
>
> We sincerely thank the reviewer for highlighting the importance of the token-ratio design and for raising the question of whether the **2:8 speech-to-reasoning ratio** is principled, robust, or potentially adaptive. We fully agree that this ratio is a critical hyperparameter in the proposed Thinking-in-Speaking paradigm.
>
> Our choice of **2:8** is motivated by the unique constraints of real-time spoken interaction. In such settings, the model must simultaneously (i) maintain continuous speech generation with tight latency budgets, and (ii) sustain an internal reasoning stream long enough to support multi-step inference. Under these constraints, 2:8 is not a hand-crafted ratio, but rather the **maximum reasoning-token allocation** that consumer-grade GPUs can stably support while producing uninterrupted real-time audio (we provide detailed throughput calculation below).
>
> To examine robustness, we conducted a controlled study using four additional token ratios: two close variants of the original configuration (**3:7** and **1:9**), and two extended-reasoning settings where the number of output tokens is fixed while reasoning tokens increase (**2:12** and **2:16**). Results are shown in **Table 2**.
>
> Table 2: Performance Comparison Across Different Speech-to-Reasoning Token Ratios on **Spoken-MQA**
>
> | Token ratio | 3:7  | 2:8  | 1:9  | 2:12 | 2:16 |
> |-------------|------|------|------|------|------|
> | Performance | 64.2 | 68.6 | 67.4 | 69.1 | 69.3 |
>
>
> The trend is consistent and clear-cut:
>
> 1.  **Compression of reasoning tokens harms performance.**
>     Moving from 2:8 to 3:7 reduces the reasoning-to-response ratio from ~4× to ~2× and causes accuracy to drop sharply to **64.2**, confirming that adequate reasoning space is essential for stable multi-step spoken inference.
>
> 2.  **Extending reasoning tokens yields only marginal gains.**
>     Moving from 1:9 → 2:12 → 2:16 improves only slightly (**67.4 → 69.1 → 69.3**).
>     This suggests that the original 2:8 already provides **sufficient reasoning length** for the model to complete its inference chain; additional reasoning tokens mainly reduce redundancy rather than introduce qualitatively new inference behavior.
>
> These results support that the 2:8 configuration lies close to a **performance plateau** where additional reasoning brings diminishing returns, while reductions lead to substantial degradation.
>
> We have also conducted a feasibility discussion regarding the **practical constraints of hardware**. Empirically, a 3090 / L40S / 4090 can generate **30.4 / 37.8 / 56.5 tokens/s** in our setup.
>
> Taking the 4090 (56.5 tokens/s) as an illustrative example:
>
> *   **2:8 ratio:**
>     Requires ≈11.3 tokens/s for speech + 45.2 tokens/s for reasoning — **well within stable real-time limits**, with no audio discontinuity observed.
> *   **2:12 ratio:**
>     Requires ≈8.1 tokens/s for speech + 48.4 tokens/s for reasoning — **already near the upper bound** of stable continuous audio generation.
> *   **2:16 ratio:**
>     Requires ≈6.3 tokens/s for speech + 50.2 tokens/s for reasoning — **approaching the hardware limit**, where minor fluctuations cause reasoning-latency spikes and audible artifacts.
> *   **1:9 ratio:**
>     Although seemingly balanced, its step structure essentially behaves like a “reasoning-dominant” configuration, requiring >100 tokens/s to sustain uninterrupted real-time dialogue — achievable only on H100-class GPUs, and **clearly impractical** for the real-time consumer-GPU settings targeted by this work.
>
> Under these constraints, 2:8 emerges as the **maximal stable configuration** that simultaneously (i) supports real-time speech generation, and (ii) provides sufficient reasoning capacity. Thus, the ratio is not arbitrary but derives directly from the interaction between model behavior and hardware-level throughput ceilings.

---

> ### Author Response · Authors · 2025-11-22
> **Rebuttal by Authors (Part-4)**
>
> > Weakness 3: The paper lacks sufficient direct comparative evidence under the same conditions to prove that "thinking-in-speaking" is superior to "think-before-speaking"—while the latter intuitively offers fuller context and stronger generalization, the former may require careful tuning (e.g., token ratio) and only partial internal reasoning capacity during speech, necessitating more robust empirical comparisons to justify the paradigm shift.
>
> We fully acknowledge the reviewer’s intuition that a **thinking-before-speaking** paradigm can provide a more complete context for response generation and may align more naturally with the training dynamics of mainstream LLMs, potentially yielding stronger results in non-real-time settings. However, our work explicitly targets **real-time spoken dialogue models**, whose requirements fundamentally differ from those of text-first systems. As discussed in Sec. 1 and Sec. 2 of the paper, end-to-end speech dialogue models were introduced precisely to avoid the substantial latency caused by **“first generate text, then synthesize speech”**, a latency stack inherently incompatible with the thinking-before-speaking paradigm.
>
> Our intention is therefore not to argue that **thinking-in-speaking** is theoretically superior to **thinking-before-speaking** in general. Rather, we aim to introduce a new paradigm tailored for real-time speech models, enabling them to maintain an internal reasoning stream while producing audio—thus satisfying both **responsiveness and answer quality** simultaneously. Empirically, we show that the **Thinking-in-Speaking** paradigm substantially reduces waiting time while achieving competitive or even better accuracy compared to **thinking-before-speaking**, which is precisely the capability gap we seek to address.

---

> ### Author Response · Authors · 2025-11-22
> **Rebuttal by Authors (Part-5)**
>
> [1] VITA-Audio: Fast Interleaved Cross-Modal Token Generation for Efficient Large Speech-Language Model. arXiv preprint.
>
> [2] Qwen2.5-Omni Technical Report. arXiv preprint.
>
> [3] VoiceBench: Benchmarking LLM-Based Voice Assistants. arXiv preprint.
>
> [4] Glm-4-voice: Towards intelligent and human-like end-to-end spoken chatbot. arXiv preprint.

---

> ### Author Response · Authors · 2025-11-28
>
> Dear Reviewer 286S,
>
> This is a gentle reminder that the discussion phase for our submission (Submission ID: 13254) will come to an end in the upcoming days. To date, we have not yet received your feedback on our rebuttal. We respectfully understand that you may have other important commitments, and your time is precious with high demands. However, we are highly eager to collaborate with you to further improve this paper—we have invested extensive efforts and made substantial revisions in response to your initial comments, aiming to address all concerns thoroughly. We sincerely hope that you find our rebuttal convincing and kindly consider revisiting your rating.
>
> We would like to express our sincere gratitude once again for your valuable time and all the constructive feedback you have provided. Your insights are crucial to enhancing the quality of our work.
>
> Thank you and best regards,
>
> The Authors of Submission 13254

---

### Note · Authors · 2026-01-06

I have read and agree with the venue's withdrawal policy on behalf of myself and my co-authors.